# Evaluating the use of blood pressure polygenic risk scores across race/ethnic background groups

Nuzulul Kurniansyah[1,41], Matthew O. Goodman[1,2,41], Alyna T. Khan[3], Jiongming Wang[4], Elena Feofanova[5], Joshua C. Bis[6], Kerri L. Wiggins[6], Jennifer E. Huffman[7], Tanika Kelly[8], Tali Elfassy[9], Xiuqing Guo[10], Walter Palmas[11], Henry J. Lin[10], Shih-Jen Hwang[12,13], Yan Gao[14], Kendra Young[15], Gregory L. Kinney[15], Jennifer A. Smith[16], Bing Yu[5], Simin Liu[17,18], Sylvia Wassertheil-Smoller[19,20], JoAnn E. Manson[2,21], Xiaofeng Zhu[22], Yii-Der Ida Chen[10], I-Te Lee[23,24], C. Charles Gu[25], Donald M. Lloyd-Jones[26], Sebastian Zöllner[4,27], Myriam Fornage[5,28], Charles Kooperberg[29], Adolfo Correa[30], Bruce M. Psaty[31,32,33,34], Donna K. Arnett[35], Carmen R. Isasi[20], Stephen S. Rich[36], Robert C. Kaplan[29,20], Susan Redline[1,2], Braxton D. Mitchell[37], Nora Franceschini[38], Daniel Levy[12,13], Jerome I. Rotter[10], Alanna C. Morrison[5] & Tamar Sofer[1,2,39,40] ✉

We assess performance and limitations of polygenic risk scores (PRSs) for multiple blood pressure (BP) phenotypes in diverse population groups. We compare "clumping-and-thresholding" (PRSice2) and LD-based (LDPred2) methods to construct PRSs from each of multiple GWAS, as well as multi-PRS approaches that sum PRSs with and without weights, including PRS-CSx. We use datasets from the MGB Biobank, TOPMed study, UK biobank, and from All of Us to train, assess, and validate PRSs in groups defined by self-reported race/ethnic background (Asian, Black, Hispanic/Latino, and White). For both SBP and DBP, the PRS-CSx based PRS, constructed as a weighted sum of PRSs developed from multiple independent GWAS, perform best across all race/ethnic backgrounds. Stratified analysis in All of Us shows that PRSs are better predictive of BP in females compared to males, individuals without obesity, and middle-aged (40-60 years) compared to older and younger individuals.

Hypertension is a major risk factor for cardiovascular disease[1,2], renal disease, and overall mortality[3], with evidence from Mendelian Randomization studies of a causal effect of blood pressure (BP) and BP-associated variants on cardiovascular disease (CVD)[4,5]. Genome-wide association studies (GWAS) of BP phenotypes, such as systolic BP (SBP), diastolic BP (DBP), pulse pressure (PP), and hypertension, have been conducted in various populations, and have identified hundreds of independent genetic variants associated with BP phenotypes[6–13].

Based on such GWAS, Polygenic Risk Scores (PRSs) have been constructed for both hypertension and BP phenotypes. These BP PRSs are associated with BP phenotypes, including rate of BP increase[14–16], and have been shown to predict incident coronary heart disease and stroke[17–22]. This demonstrates that BP PRSs may identify at-risk individuals long before they develop elevated BP. Polygenic scores for various health outcomes are now being studied for integration in routine medical care[23,24]. A fundamental question is whether and at what

performance threshold to adopt a BP PRS clinically, using it to guide early intervention via drug treatment or lifestyle changes for those at greatest genetic risk of progression to hypertension or adverse CVD outcomes. Answering this question requires assessment of research inequities due to lagging PRS performance in non-European populations[25–27]. This inequity may translate to healthcare inequity if PRSs trained in European/White populations pass a threshold of clinical utility first, and are of primary benefit to White individuals in identifying patients at risk of future disease[26].

Populations with diverse global ancestries in the US, especially Hispanic/Latino and Black Americans, are at increased risk of elevated BP[28,29], yet genetic studies have been unevenly and inequitably distributed globally, yielding few GWAS of BP in relevant populations. It is therefore important to evaluate PRS performance in multi-ethnic populations, including admixed populations, such as those represented in Trans-Omics in Precision Medicine Initiative (TOPMed)[25]. Previously, we studied the generalization of PRSs for BP traits from White to Hispanic/Latino and Black Americans[30]. We showed that, even when utilizing GWAS results from Hispanic/Latino participants to select variants into the PRS or to compute weights, BP PRSs based primarily on GWAS results in White individuals had poor performance in Hispanic/Latino individuals. Furthermore, PRSs based on GWAS results in White participants were only weakly associated with BP phenotypes in Black Americans. Since then, GWAS based on larger and more diverse study populations have been published, and additional methods to take advantage of multiple GWAS in multi-ethnic PRSs have been proposed, including GWAS meta-analysis and weighted-sum approaches. Using GWAS meta-analysis as the basis for PRS has been criticized for potentially obscuring population-specific genetic effects, and presents the further question of choosing a single reference-panel population from which to compute linkage disequilibrium (LD, i.e. correlation between SNPs) to select or adjust the marginal SNP effects of independent blocks of SNPs. The recently proposed multiPRS[31] and metaGRS[32] use a weighted-sum of previously validated and optimized PRSs. In keeping with this general approach, PRS-CSx[33] has been developed to compute ancestry-specific PRSs based on GWAS performed on populations of distinct genetic ancestries, followed by summations of the ancestry-specific PRSs.

Environment and lifestyle, as well as gene-by-environment interactions, have been implicated in the BP variability that remains beyond what current PRSs explain[34]. Individuals of diverse backgrounds tend to experience different environments. Cultural environment may influence diet; sociocultural environment may lead to differential social adversity, including effects of economic disadvantage and structural racism[35], which may in turn cause or accentuate exposures such as to stress and environmental pollutants, and impose limits on leisure-time physical activity, nutrition options, or access to healthcare. These environmental factors may interact with genetic factors, including unknown or under-researched genetic factors that are unique to or predominant in certain populations[36]. Thus, while PRSs may have different performance across populations due to genetic architecture factors such as LD structures, PRS associations with their phenotypes may also differ across population groups such as those defined by self-reported race/ethnicity due to various gene-environment interactions. Such environmental exposures may not be consistently measured at scale, i.e., using the same instruments across multiple studies.

We sought to evaluate multi-ethnic performance of BP PRSs constructed using multiple GWAS and methods. Our study population included individuals from multiple diverse backgrounds in several large biobank and consortia studies: Asian or Asian American (Asian), Black or African American/British (Black), Latino or Hispanic American (Hispanic/Latino), and White or European American (White). While these groups correspond to socially-defined racial/ethnic identities, which are social constructs that may shift according to context and

change over time[37,38], these constructs also imply differing distributions of continental genetic ancestry and admixture[39,40]. Acknowledging both the social and genetic aspects of these identities, we describe them as race/ethnic "backgrounds". We sought to better understand how PRS performance varies according to race/ethnic background, thereby shedding light on how the use of standard PRS methodologies for BP phenotypes may potentially impact health equity. This approach complements ongoing efforts to refine PRS methodology, as well as scientific initiatives to fund research in diverse genetic backgrounds and admixed populations. Figure 1 provides an overview of the PRS preparation, as well as hypotheses underlying the analyses.

## Results

Characteristics of TOPMed participants, UK Biobank Black participants, MGB Biobank, and All of Us participants are provided in Supplementary Tables 1, 2, 3, and 4. Supplementary Figs. 1 and 2 provide the observed phenotypic and residual variances of SBP and DBP stratified by study and race/ethnic background, where residuals were obtained from models using covariates only (without PRSs). Residual variance, the denominator in the proportion variance explained statistic, varied substantially across cohorts, and was generally lowest in White, higher in Hispanic/Latino and Asian and highest in Black participants, by up to 25% over White participants. Estimated global proportions of continental genetic ancestries of TOPMed-BP participants are visualized in Supplementary Fig. 3. One can see that Black participants are admixed with primarily African ancestries and some European ancestries, and Hispanic/Latino participants are admixed with substantial European, Amerindian, and African ancestries. We also see substantial fractions of Middle Eastern ancestries, potentially in error due to the similarity between European and Middle Eastern ancestries.

### PRS performance by race/ethnic backgrounds

PVE for SBP and DBP PRSs across race/ethnic background groups are visualized in Figs. 2 and 3 respectively, and the data behind these figures are provided in Supplementary Data 1 and Supplementary Data 2. The number of participants in each of the PRS analyses is provided in Supplementary Table 5. In all background groups, sums of PRSs performed better than single GWAS PRSs, with the best performing PRS being weighted sums of PRS-CSx ancestry-specific PRSs. Note that both PRS-CSx1 and PRS-CSx2 outperformed each other in some settings. Thus, we chose PRS-CSx2 (which used COGENT to represent the African ancestry component) as the best performing PRS because it had the highest PVE in a multi-ethnic analysis combining all race/ethnic groups: for SBP, PRS-CSx2 PVE was 6.0% compared to 5.4% for PCS-CSx1, and for DBP PRS-CSx2 PVE was 5.8% compared with 4.9% for PRS-CSx1. Across race/ethnic backgrounds, the PVE of PRS-CSx2 ranged from 8.0% (SBP) and 7.8% (DBP) in White to 3.5% (SBP) and 3.1% (DBP) in Black individuals. In the primary analysis, where the TOPMed-BP dataset was used as a reference panel, the LDPred2 PRS based on UKBB + ICBP GWAS of European ancestry individuals outperformed the other single-GWAS PRSs in both White and Hispanic/Latino participants, while LDPred2 MVP PRS performed better in Black participants. In Asian participants, the LDPred2 MVP-based PRS performed best for SBP whereas the LDPred2 UKB-ICBP performed best for DBP. In the GitHub repository, we provide estimated PVEs for all compared PRSs and race/ethnic background groups, as well as corresponding PRS effect sizes, per standard deviation in TOPMed-BP dataset, as estimated in the combined TOPMed-BP and UKB Black dataset. Means and SDs of the PRS-CSx2 PRSs and their component PRSs based on the TOPMed-BP dataset are provided in Supplementary Table 6 and 7. Similarly, MGB Biobank-trained PRS summation weights are provided for all relevant PRSs in the GitHub repository, and for the best performing PRSs in Supplementary Table 8.

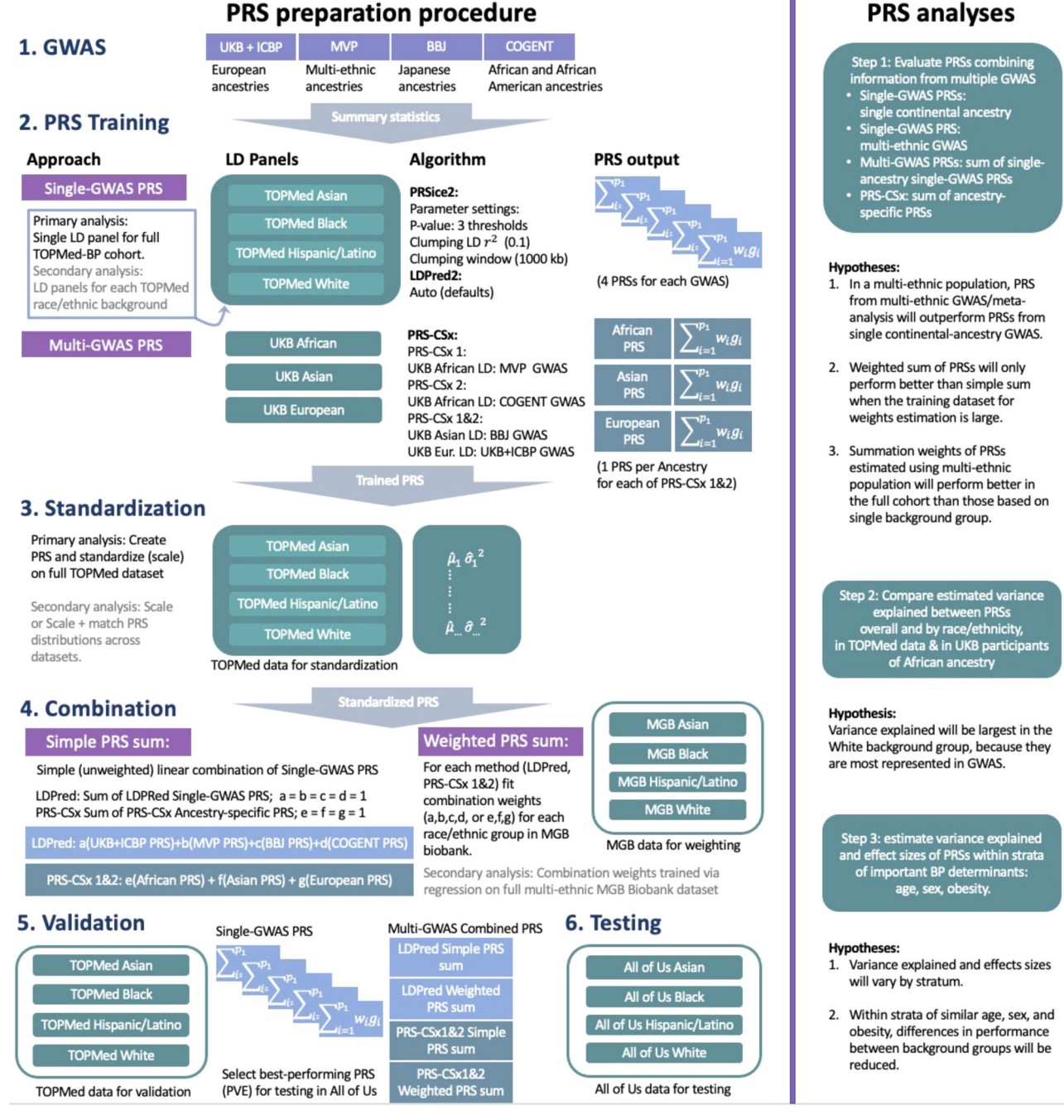

**Fig. 1 | PRS preparation and analyses steps.** The left panel shows the GWAS summary statistics, LD reference panels, computational procedures and PRS standardization involved in PRS preparation. The right panel describes the major analytic steps and their associated hypotheses. BBJ Biobank Japan, COGENT Continental Origins and Genetic Epidemiology Network, GWAS genome-wide association study, LD linkage disequilibrium, LDPred Bayesian PRS method (LDPred2 is a specific software implementation of the LDPred algorithm), MGB Mass General Brigham, MVP Million Veteran Program, PRS-CSx continuous shrinkage cross-population PRS method, PRS polygenic risk score, PRSice2 software for computing PRS based on the clumping & thresholding methodology, TOPMed Trans-Omics for Precision Medicine, UKBB + ICBP United Kingdom biobank and the International Consortium of Blood Pressure.

Supplementary Figs. 4 and 5 describe a series of secondary analyses relating to PRS construction methods. Briefly, training weights for PRS combination methods using background-specific analysis in MGB Biobank was not superior to training weights using the complete, multi-ethnic MGB dataset (Supplementary Fig. 4), and using reference panel based on individuals with the same continental ancestry as the individuals used for GWAS had better performance than using a multi-ethnic reference panel (Supplementary Fig. 5).

Example comparison of PRS distributions across datasets is provided in Supplementary Figs. 6 and 7. Supplementary Figs. 8 and 9 demonstrate the potential impact of PRS scaling and PRS scaling + matching approaches between the dataset used for training summation weights (MGB Biobank) and the dataset used for PRS evaluation (TOPMed-BP and UKBB Black) on PRS performance in the evaluation dataset. In brief, without explicitly matching PRS distributions across datasets, there are clear differences in their distributions between datasets. However, different scaling and scaling + matching approaches resulted in minimal differences in PVE in TOPMed-BP. In another sensitivity analysis we raised the values of SBP and DBP in MGB Biobank individuals with history of antihypertensive medication use. The

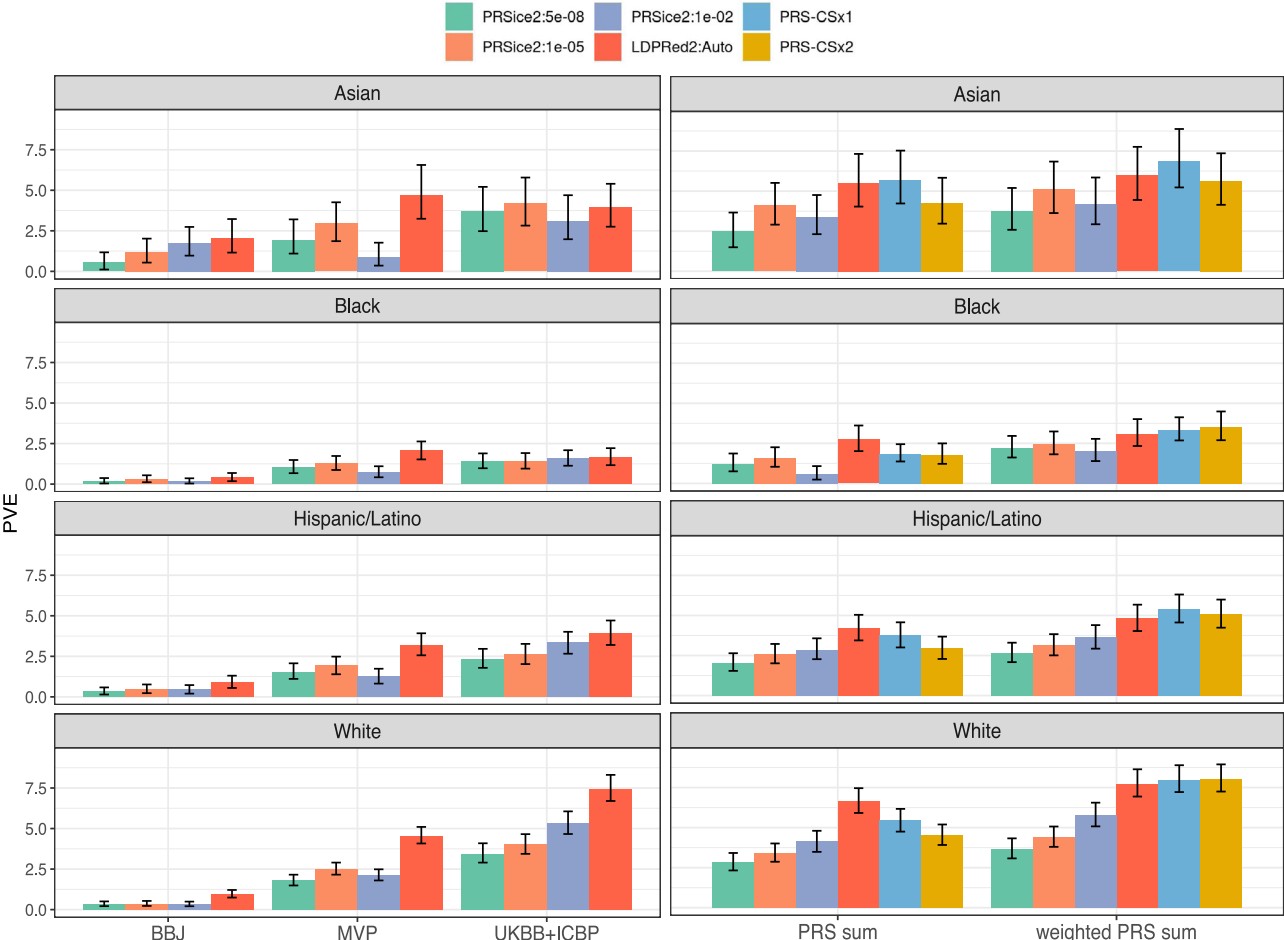

**Fig. 2 | SBP variance explained by compared SBP PRSs.** Estimated variance explained (PVE) by all compared SBP PRSs in the TOPMed-BP datasets (and UKBB Black individuals), stratified by race/ethnic background. The height of each bar represents the estimated PVE from association analysis of the PRS with SBP in the given stratum. Intervals represent the 95% confidence intervals based on the 2.5% and 97.5% distribution percentiles from bootstrap performed using unrelated individuals. The left column corresponds to PRSs constructed based on single GWAS, and the right column corresponds to PRS summation approaches. PRS-CSx1 and PRS-CSx2 refer to implementations of PRS-CS in which either MVP (PRS-CSx1) or COGENT (PRS-CSx2) GWAS summary statistics were used in place of an African

ancestry GWAS. The sample sizes used in each of the analyses represented by the different bars in the figure are provided in Supplementary Table 5. BBJ Biobank Japan, GWAS genome-wide association study, LDPred Bayesian PRS method (LDPred2 denotes a specific software implementation of the LDPred algorithm), MVP Million Veteran Program, PRS-CSx continuous shrinkage cross-population PRS method, PRS polygenic risk score, PRSice2 software for computing PRS based on the clumping & thresholding methodology, PVE percent variant explained, SBP systolic blood pressure, TOPMed Trans-Omics for Precision Medicine, UKBB + ICBP United Kingdom biobank and the International Consortium of Blood Pressure.

performance of weighted summations of PRSs in TOPMed remained similar to the analogous performance in the primary analysis (Supplementary Fig. 10).

## PRS distribution is affected by population structure of the GWAS population

Supplementary Figs. 11 and 12 describe the distributions of the highest performing single GWAS PRS and multi-PRS, in each TOPMed race/ethnic background group, and by groups defined by genetic ancestry. As is well known, PRS distributions differ across background groups, due to differences in allele frequencies. Supplementary Figs. 13 and 14 visualizes the patterns of associations between ancestry-specific allele frequencies and SNP effect sizes using SNPs from the four GWAS summary statistics underlying our analyses. For the population that most closely matches the ancestry distribution of the underlying GWAS, the figure recapitulates the known results that common SNPs have weaker effect sizes compared to rare SNPs. This can be seen, for example, in the "U"-shaped curve observed when plotting SNP effects from the UKBB + ICBP GWAS and European-specific allele frequencies from TOPMed. However, the same organized pattern is not observed when

considering allele frequencies of a differing ancestral population, where the allele frequencies have essentially been shuffled for some variants, resulting in a wider distribution of effect sizes at a given frequency. For MVP GWAS, which is multi-ethnic, the "U"-shaped pattern is roughly observed for European and African ancestry-specific frequencies, suggesting that multi-ethnic analysis may better localize signal SNPs.

## Estimated effect sizes and PVE of BP PRS stratified by major BP determinants in All of Us

We further validated the top-performing PRS (PRS-CSx2) in the All of Us dataset. Supplementary Table 4 characterizes the study population. Over than 20% of All of Us participants use antihypertensive medications. Figure 4 provides the estimated effect sizes and PVEs in each race/ethnic background, demonstrating similar pattern to the TOPMed-BP dataset, but lower PVEs, and slightly higher PRS performance in the Hispanic/Latino group compared to the White group (the data behind the figure is provided in Supplementary Data 3). Supplementary Fig. 15 describes similar results, from analysis stratified by groups defined by a combination of self-reported race/ethnicity and genetic similarity. Results across corresponding groups between Fig. 4

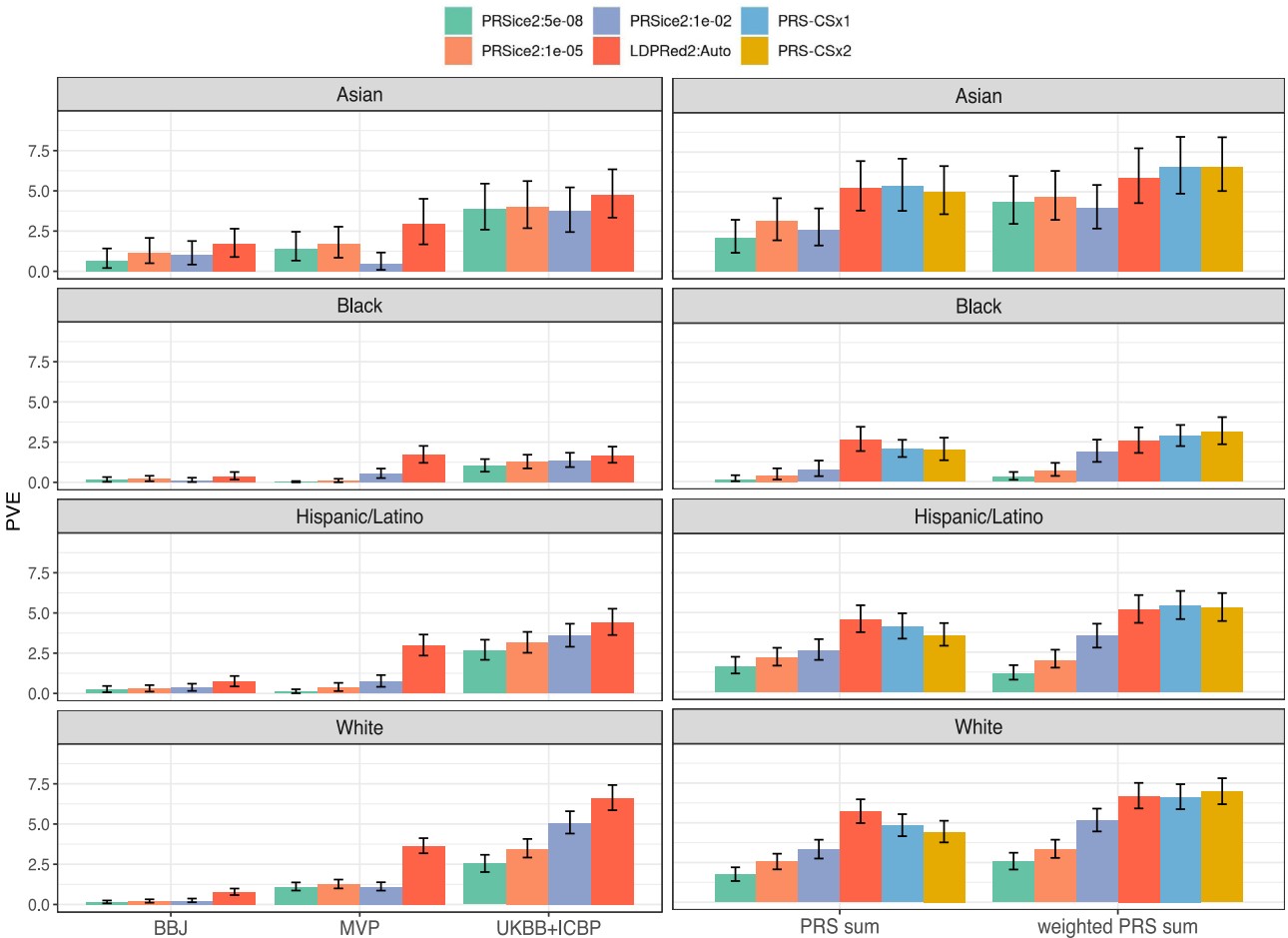

**Fig. 3 | DBP variance explained by compared DBP PRSs.** Estimated variance explained by all compared DBP PRSs in the TOPMed-BP datasets (and UKBB Black individuals), stratified by race/ethnic background. The height of each bar represents the estimated PVE from association analysis of the PRS with SBP in the given stratum. Intervals represent the 95% confidence intervals based on the 2.5% and 97.5% distribution percentiles from bootstrap performed using unrelated individuals. The left column corresponds to PRSs constructed based on single GWAS, and the right column corresponds to PRS summation approaches. PRS-CSx1 and PRS-CSx2 refer to implementations of PRS-CS in which either MVP (PRS-CSx1) or COGENT (PRS-CSx2) GWAS summary statistics were used in place of an African

ancestry GWAS. The sample sizes used in each of the analyses represented by the different bars in the figure are provided in Supplementary Table 5. BBJ Biobank Japan, DBP diastolic blood pressure, GWAS genome-wide association study, LDPred Bayesian PRS method (LDPred2 denotes a specific software implementation of the LDPred algorithm), MVP Million Veteran Program, PRS-CSx continuous shrinkage cross-population PRS method, PRS polygenic risk score, PRSice2 software for computing PRS based on the clumping & thresholding methodology, PVE percent variance explained, TOPMed Trans-Omics for Precision Medicine, UKBB + ICBP United Kingdom biobank and the International Consortium of Blood Pressure.

and Supplementary Fig. 15 are fairly similar and it is not clear that grouping individuals while accounting for genetic similarity results in improved PRS performance. Supplementary Figs. 16 and 17 provide the corresponding results from analyses stratified by hypertensive medication use, showing that the associations between BP PRSs and BP traits remained about the same in non-medication users as in the primary analysis, but with slightly lower effect sizes. PRS associations are weaker, but still present, in participants who use antihypertensives. When stratifying by major BP determinants (age, sex, and BMI), performance and PRS effect sizes varied by strata. The patterns were largely similar across background groups, so for simplicity, Fig. 5 focuses on the multi-ethnic analysis while Supplementary Figs. 18–21 provide results by race/ethnic background. One can see that SBP PRS associations are stronger in females (PVE 3.4%, beta = 3.8) compared to males (PVE 2.5%, beta = 3.3), in non-obese (PVE 3.5%, beta = 3.7) compared to obese (PVE 2.3%, beta = 3.3) individuals, and in individuals aged 41-60 (PVE 3.5%, beta = 4.1) compared to those younger (PVE 2.6%, beta = 2.8) or older (PVE 2.7%, beta = 3.6). The PRS-CSx2 DBP PRS exhibits similar patterns. These stratified performance patterns seen in the multi-ethnic analysis most closely match those seen in the White

participants, whereas in the Supplement we show that, while broadly similar, for other groups some patterns were changed or attenuated. For example, male/female and obese/non-obese performance were generally more similar, while relative performance was better in younger (age < 40) Black participants for SBP and DBP, and younger Asian participants for DBP. The data behind Fig. 5 is provided in Supplementary Data 4. Multi-ethnic analysis stratified by both medication use and BP determinants follow similar patterns to the multi-ethnic analysis: among non-medication users (Supplementary Fig. 22) the associations remain similar to the primary analysis results in Fig. 5, but with somewhat attenuated effect size estimates, while in medication users (Supplementary Fig. 23) the associations are weaker. Notable difference is that among medication users, BP PRS PVE is highest in the stratum of individuals ages 40 or less, unlike in those who do not use medications.

## BP PRS are associated with prevalent clinical outcomes in All of Us

We also tested the association of SBP and DBP PRS with prevalent clinical outcomes in All of Us, and compared them with our

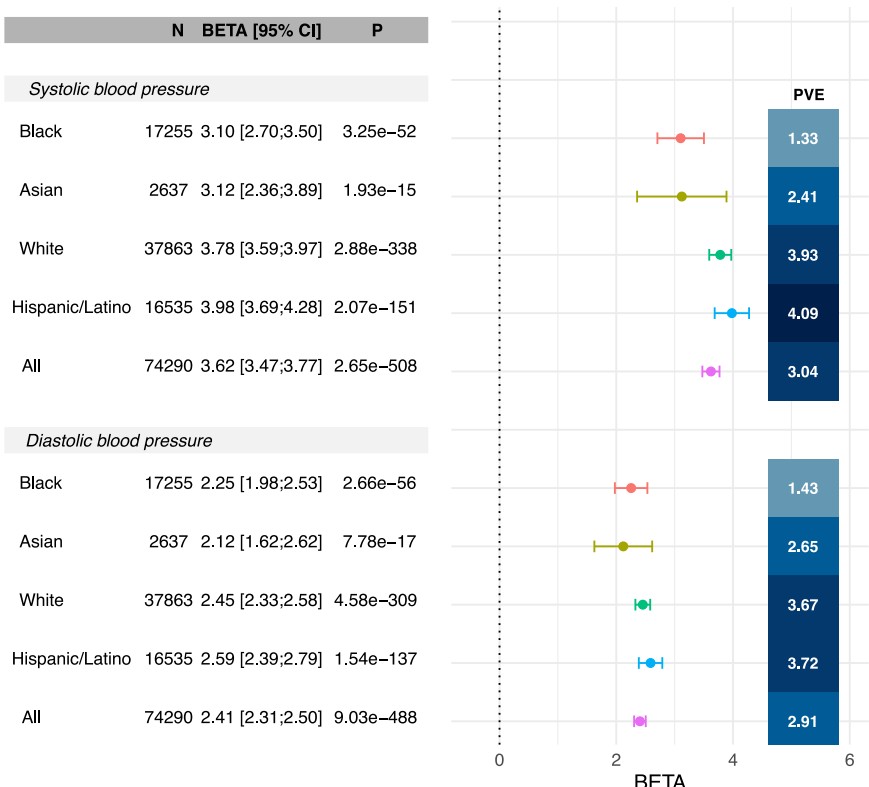

**Fig. 4 | Association of SBP and DBP PRS with BP measures stratified by race/ethnic background in the All of Us dataset.** The figure visualizes the estimated associations (betas in units of mmHg per 1 SD increase in the PRS, standardized according to TOPMed; provided numerically and visualized as points in the forest plot) and 95% confidence intervals (CIs; provided both numerically and visualized as the intervals in the forest plot) of the best performing SBP and DBP PRSs as selected in the TOPMed dataset (PRS-CSx2), in All of Us, stratified by race/ethnic

background. The figure also provides the sample sizes, in each stratum, association *p*-values computed based on a two-sided 1-degree of freedom Wald test, and PVEs. Analyses were adjusted for age, sex at birth, BMI, self-reported race/ethnicity, and the first 10 PCs of genetic data. DBP diastolic blood pressure, PRS-CSx continuous shrinkage cross-population PRS method, PRS polygenic risk score, PVE percent variance explained, SBP systolic blood pressure, TOPMed Trans-Omics for Precision Medicine.

previously-developed hypertension PRS (HTN-PRS) which combined PRSs based each on a single multi-ethnic SBP, DBP, and hypertension GWAS using clump-and-threshold PRSice methodology, as well with a newly developed weighted sum of the PRS-CSx2 SBP and DBP PRSs. Results are provided in Supplementary Fig. 24, and further Supplementary Figs. 25 and 26 report results from analyses stratified by antihypertensive medication use. In primary analysis, all PRSs were associated with the outcomes. AUC of the PRS associations with hypertension were similar and equal 0.71 for SBP, DBP, and SBP and DBP weighted sum, and slightly lower (0.70) for HTN-PRS. SBP and DBP PRS weighted sum had the strongest association *p*-value with hypertension. Similarly for other outcomes, including type 2 diabetes, chronic kidney disease, coronary artery disease, atrial fibrillation, and heart failure, associations were fairly similar across PRSs. Results were inconsistent in analyses stratified by antihypertensive use. For example, HTN-PRS was associated with multiple outcomes (type 2 diabetes, chronic kidney disease, and heart failure) among medication users, while other PRSs were not. However, both SBP and DBP PRS and their weighted sum were (weakly) associated with chronic kidney disease among non-medication users, while HTN-PRS was not. Given the width of the confidence intervals, we do not have sufficient power to make strong conclusions.

## Discussion

We performed an in-depth evaluation of BP PRS associations with BP phenotypes in diverse populations, using varying methodologies reflective of current PRS construction practices, in conjunction with the largest available single-ancestry and multi-ethnic GWAS. This

manuscript expands upon prior work in its focus on multi-ethnic, multi-ancestry populations. Similar to our recently published paper about a multi-ethnic HTN-PRS[17], our intent was to develop PRSs that are useful across individuals regardless of both their genetic make-up and their race/ethnic identity, while acknowledging that PRS performance may be related to these factors. In prior work, we considered summary statistics from multi-ethnic GWASs, trained a HTN-PRS summing trait-specific SBP, DBP, and hypertension PRS without weights, and used TOPMed as a primary dataset, and included a longitudinal and incidence analyses. Here, we developed PRSs based on multiple PRS methods and multiple GWAS summary statistics, evaluated them across population strata, and utilized multiple datasets to further assess PRS performance by strata defined by important determinants of BP.

Our hypothesis that multi-ethnic GWAS can yield PRS with superior performance to single-ancestry GWAS and meta-analysis was partially confirmed. For White and, perhaps contrary to expectation, Hispanic participants, the highest-performing single-GWAS PRS was obtained using LDPred2 along with the UKBB + ICBP GWAS consisting solely of European ancestry individuals. For Black and Asian participants, the best-performing single-GWAS PRS was based on the multi-ethnic MVP GWAS. PVE performance differed substantially by race/ethnic background and was lowest in Black participants, corresponding to the smaller GWAS sample size for African continental ancestry. The highest performing PRS for each race/ethnic background, among single- and multi-GWAS PRSs, was the PRS-CSx1, using fixed weights across background groups trained in the MGB Biobank). While we did not report results for PRS based on meta-analysis of the four GWAS

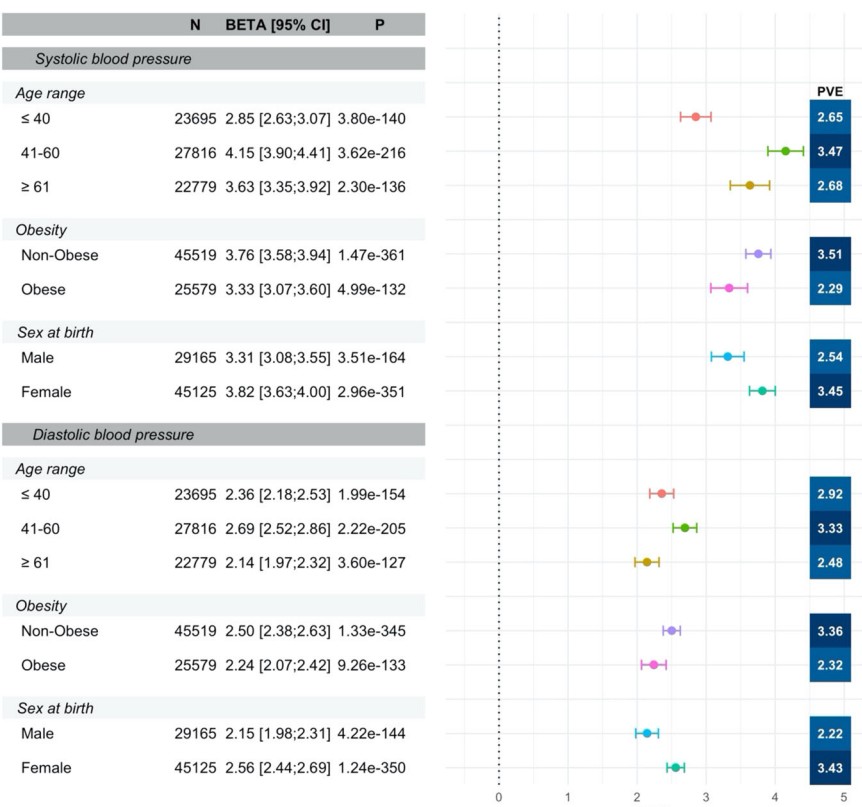

**Fig. 5 | Associations of BP PRSs with BP measures stratified by age range, obesity and sex at birth in the All of Us dataset.** The figure visualizes the estimated associations (betas in units of mmHg per 1 SD increase in the PRS, standardized according to TOPMed; provided numerically and visualized as points in the forest plot) and 95% confidence intervals (CIs; provided numerically and visualized as the intervals in the forest plot) of the best performing SBP and DBP PRSs as selected in the TOPMed dataset (PRS-CSx2) with their respective phenotypes in All of Us, stratified by major hypertension risk factors. The figure also provides the sample sizes, in each stratum, association *p*-values computed based on a two-sided 1-degree of freedom Wald test, and PVEs. Analyses were adjusted for age, sex at birth, BMI, self-reported race/ethnicity, and the first 10 PCs of genetic data. DBP diastolic blood pressure, PRS-CSx continuous shrinkage cross-population PRS method, PRS polygenic risk score, PVE percent variance explained, SBP systolic blood pressure, TOPMed Trans-Omics for Precision Medicine.

contributing to this manuscript due to differences in outcome scales, multi-PRS approaches provide a useful alternative as they do not depend on scaling of the outcome in individual GWAS. It is a topic of future work to meta-analyze summary statistics from different GWAS by potentially rescaling the effect sizes to bring them to the same scale.

Importantly for clinical adoption of PRSs for BP and other important phenotypes, current PRS performance differs by race/ethnic background in a systematic way. Achieved performance largely tracks available sample size from underlying GWAS, again emphasizing the need for additional recruitment of GWAS participants from underrepresented groups. This is in line with other reports[25,41,42]. Secondary analyses suggest that relative PVE performance across race/ethnic background groups may also be driven in part by differences in BP distributions, covariate distributions and effects, and age-dependent BP trajectories.

PRS distribution diverges by race/ethnic background. These effects are attributable to differential patterns of allele distribution (frequency and LD) across the groups with respect to the fixed PRS estimated allelic effect weights, and can potentially bias PRS associations and interpretations, and their interpretation in multi-ethnic studies. These ancestry-specific distributional effects have been consistently noted before[17,25]. PRS distribution changes by genetic ancestry highlight the difficulty in using PRS to determine disease risk in admixed individuals.

Consistent with expectation, PRS based on the largest GWAS, UKBB + ICBP, trained in individuals of European ancestry, outperformed other PRS and achieved the highest PVE performance overall, when evaluated on White participants in the TOPMed test cohort. Surprisingly, performance of the UKBB + ICBP PRS in Hispanic/Latino participants was roughly equivalent, perhaps due to large proportion of European ancestry in Hispanic/Latino admixed populations[39].

Perhaps more surprisingly, the performance of the UKBB + ICBP PRS was also relatively high among Asian and Black participants, compared with other alternatives, exceeding the performance of BBJ (Japanese population), and roughly equaling the performance of MVP (multi-ethnic population). This suggests the PRS performance benefits to both the target ancestry and other ancestries of large-sample underlying GWAS analyzing, here about ~700 K individuals, from a common continental ancestry. Nonetheless, for all groups, performance of multi-GWAS PRS exceeded that of the best-performing based single-ancestry GWAS. However, current methodology to borrow strength across GWAS of different continental ancestries is not capable of making up the difference and equalizing performance for underrepresented background groups. This points to the likely persistence of performance inequalities across race/ethnic backgrounds, unless funding is allocated to equalize representation of continental ancestries.

To better understand the role of underlying phenotypic variance and GxE effects in quantifying PRS performance, we studied PRS performance by strata of important BP determinants in the validation dataset All of Us. This follows reports by others demonstrating that PRS performance may differ across population segments defined by environmental and clinical factors[43]. Indeed, PRS performance varied

by age, sex, and obesity strata, affirming the need to account for these factors as well as other sources of heterogeneity by cohort when evaluating and utilizing PRS to predict phenotypes that have strong age and other clinical and environmental dependencies. In supplemental analyses, the above-described stratified PRS performance patterns were seen more clearly in White and Hispanic backgrounds than in Asian and Black backgrounds, again suggesting a role for more complex environmental and GxE effects.

This study has several notable strengths. We have attempted to characterize the performance of BP PRS developed based on largest available GWAS summary statistics and modern analytic methods for studies involving multiple ancestries and reported backgrounds. We specifically compared performance of PRS by self-reported race/ethnic background, rather than calculated genetic ancestry because at a time when PRS are coming closer to clinical use, we focus on investigating whether PRS perpetuate a socially constructed racial bias and highlight equity issues for BP PRS related to these social identities. Specifically, the use of race/ethnic background groups reflects potential clinical application, where PRS may be trained and adopted based on reported background rather than more complex models of genetic ancestry and admixture patterns. We identified and produced PRS from a variety of large-scale GWAS in diverse populations to enhance PRS to reflect the unique genetic architecture of BP in the various genetic ancestries represented in the reported backgrounds. We followed standard practices for clumping and thresholding based PRS construction as well as newer Bayesian model-based PRS, and investigated a few approaches to address multi-ancestry populations, including multi-ancestry GWAS and weighted and unweighted PRS summations, where we used the independent MGB Biobank dataset to estimate summation weights. However, we note that the MGB Biobank is limited in being a biobank population, that tends to have more measures related to hospital-based encounters (compared to cohort studies that have baseline and periodic surveys of all individuals). Thus, both BP phenotypes and medication data are "noisy", and better PRS combination weights could potentially be developed. We also investigated the effect of background-specific vs multi-ancestry LD reference panels. We followed TOPMed best practices[44] for genetic analyses involving diverse ancestries and reported social racial groupings. Specifically, we accounted for differential intercepts and heteroskedasticity by reported race/ethnic background. We tested the predictive capability of each of these PRS in the portion of the TOPMed-BP dataset that did not overlap with the training GWAS, and finally validated the top performing PRSs in a the All of Us, multi-ethnic, validation study. In secondary analysis we evaluated PRS associations with prevalent clinical outcomes in All of Us, with further stratification by antihypertensive medication use. While this analysis is useful in that it provides evidence for the potential use of BP PRSs in capturing risk that relates to specific outcomes, it is limited in that we did not evaluate associations with incident outcomes. Also, in future work, phenotypes and study samples should be further elaborated with specific, pre-defined, relevant exclusion and inclusion criteria to yield useful insights for future application in clinical care.

This work highlights challenges to address in developing PRSs for diverse populations. First, consistent with other work, these results underline the clear need to recruit to genetic studies individuals who, combined, fully represent the diversity of the underlying population, including oversampling to achieve adequate power. Second, the results imply a continued need to evaluate and improve methods to account for the ancestry-dependent distribution of PRS. Consequences on PRS include bias, miscalibration, and loss of precision. To better correct these biases, it would be useful to compare conditioning on PCs to other methods, e.g. those based on proportion ancestry for admixed populations. Third, more data collection and methodology are needed to develop PRSs that are useful in diverse populations by appropriately accounting for environmental factors that may vary in distribution by background group and may be involved in GxE interactions, in addition to genetic ancestry distributions. Because race/ethnic background tend to be a proxy for social adversity and racial discrimination, as well as additional environment and lifestyle factors, training and calibration of PRS by reported race/ethnicity could induce bias or perpetuate discrimination, including in individuals identifying with one group but who are relative outliers in their genetic ancestral make-up, or lead to situations where GxE effects driven by group-specific exposures are interpreted as immutable ancestry-specific genetic effects. On the other hand, such grouping may be useful for prediction and early intervention precisely because typical exposures across the lifespan, that modify genetic effects, may be more similar within socially-constructed groups. Ideally risk models would disaggregate objective properties of genetic ancestry (admixture, underlying ancestry-specific allele frequency and LD structures) from social adversity and other environmental factors that vary by background group.

In summary, we found that combining multiple PRSs based on different GWAS is useful, and PRS-CSx software that computes ancestry-specific PRS, which are then combined, leads to strong PRSs that perform well across race/ethnic background strata. Our work also suggests that to improve PRS performance across race/ethnic backgrounds requires understanding all factors driving performance: not only PRS characteristics driven by population structure, but also cohort effects, differing distributions of age and other covariates by race/ethnic background, as well as GxE effects, including those that are due to genetic or environmental factors more prevalent in certain backgrounds. Construction and evaluation of BP PRS risk models for clinical use may need to appropriately account for all these factors to avoid perpetuating research and healthcare inequity.

## Methods

This analysis used multiple datasets. The primary evaluation dataset is the TOPMed-BP, which was used for developing some of the PRSs, assessing PRSs, and study of genetic ancestry-related patterns in PRSs and their associations. All PRSs were standardized according to means and standard deviations (SDs) estimated in the TOPMed-BP dataset. Second, we used the Mass Genetic Brigham (MGB) Biobank to train weights for multi-PRS combinations (weighted sums of PRSs). Third, we used Black individuals from the UK Biobank to complement TOPMed individuals self-reported as Black, because the available sample size was small after eliminating overlap with one of the GWAS summary statistics used. Finally, we used the All of Us dataset for final validation of PRS performance and effect size estimation in an independent dataset of diverse individuals from the U.S. All of Us validation was applied only on the highest performing PRSs in the main evaluation dataset. UK Biobank, MGB Biobank, and All of Us methods are provided in Supplementary Note 1. Below we describe the TOPMed-BP dataset; information about other datasets is provide in the Supplementary Note 2.

### The TOPMed-BP dataset

The TOPMed-BP dataset includes 62,501 individuals from 15 parent population-based studies. Individuals are categorized into four race/ethnic "background" identity groups, based on self- or study-identification. We used BP phenotypes that were harmonized by the TOPMed Data Coordinating Center (DCC), according to its published methodology[45] based on data downloaded from dbGaP. For each cohort, we generally used BP phenotypes, covariates, and medication data from the first available exam, to maximize the sample size. We studied SBP and DBP. As accepted elsewhere in genetic studies of BP phenotypes, SBP and DBP values were raised by 15 mmHg and 10 mmHg, respectively, in individuals using antihypertensive medications[46].

**Table 1 | Training GWAS used as references for PRS construction**

| GWAS name | Reference | Trait and sample size | Population | Overlap with TOPMed |
|---|---|---|---|---|
| MVP | PMID: 30578418[6] | SBP $n = 318,492$ DBP $n = 318,891$ | Multi-ethnic (69.1% non-Hispanic White, 18.8% non-Hispanic Black, 6.7% Hispanic, 0.77% non-Hispanic Asian and 0.85% non-Hispanic Native American | None |
| BBJ | PMID:29403010[57] | SBP $n = 136,597$ DBP $n = 136,615$ | Japanese | None |
| UKBB + ICBP | PMID:30224653[7] | SBP $n = 757,601$ DBP $n = 757,601$ | European and European American | 13,516 |
| COGENT | PMID: 28498854[9] | SBP $n = 31,155$ DBP $n = 31,155$ | African and African American | 9708 |

*UKBB + ICBP* United Kingdom biobank and the International Consortium of Blood Pressure, *MVP* Million Veteran Program, *BBJ* Biobank Japan, *COGENT* Continental Origins and Genetic Epidemiology Network.

## Genome sequencing

We used genetic data from whole genome sequencing via the TOPMed program[47] freeze 8 release. The TOPMed Data Coordinating Center constructed a kinship matrix estimating recent genetic relatedness, the corresponding sparse kinship matrix, where values were set to zero when the genetic relationship was estimated to be more distant than 4th degree relatedness, as well as providing genetic principal components (PCs), using the PC-Relate algorithm[48]. Information about genome sequencing, allele calling, and quality control in TOPMed is provided in https://www.nhlbiwgs.org/topmed-whole-genome-sequencing-methods-freeze-8. In Supplementary Note 1, we describe genetic ancestry inference for TOPMed participants. We used the inferred genetic ancestry for secondary analyses using genetic ancestry-defined groups, and to apply the GAFA algorithm[49] and compute ancestry-specific allele frequencies.

## Training GWAS of BP phenotypes

Table 1 provides information about GWAS that we used for summary statistics to construct PRS. The largest GWAS, UKBB + ICBP, was of 757,601 individuals of European ancestry, and the smallest GWAS, COGENT, was of 31,155 individuals of African ancestry and African and European admixture. All GWAS in Table 1 are non-overlapping with each other. Because COGENT is relatively small for PRS construction, we only used it when combining information from the four GWAS in multi-PRS approaches. We did not meta-analyze results across GWAS, because effect size estimates were on different scales, due to transformation of outcomes in the different GWAS. Both UKBB + ICBP and COGENT have overlapping individuals with some of our TOPMed cohorts (Table 1). We excluded such individuals from TOPMed-BP PRS assessment when developing PRSs using information from these GWAS. Supplementary Table 1 in the Supplementary Information provides sample sizes used when constructing PRS based on each of the GWAS summary statistics used and the multi-PRS methods.

## Quality control on summary statistics

We excluded from consideration SNPs with MAF < 0.01 in the TOPMed-BP dataset and/or in the discovery GWAS (shown in Table 1), as well as SNPs that did not pass quality control filters in TOPMed and SNPs with missing values in at least 1% of TOPMed-BP individuals. Note that we did not directly filter SNPs based on a Hardy-Weinberg Equilibrium test in TOPMed, as other quality control filters already address Mendelian inconsistencies (see link in the above subsection for quality control measures in TOPMed).

## PRS construction based on a single GWAS

We constructed PRSs using the clump-and-threshold method implemented in PRSice2 software version 2.3.1.e[50] and using the linkage-disequilibrium based approach implemented in LDPred2[51,52], implemented in the bigsnpr v1.0.11 R package. Following quality control, we

re-coded the variant positions (via the UCSC hg19 to hg38 chain file) and alleles (using PRSice software to perform allele flipping and remove ambiguous alleles) to match those in the TOPMed data (build hg38). Then, we applied PRSice using data from each of the GWAS in Table 1 other than COGENT. We used *p*-value thresholds of $5 \times 10^{-8}$, $1 \times 10^{-5}$, and $1 \times 10^{-2}$ for SNP inclusion. For clumping, we used the combined TOPMed-BP dataset as a reference panel for Linkage Disequilibrium (LD), and set clumping parameters to $R^2 = 0.1$ and distance = 1000 Kb, so that if a variant was selected into the PRS, other variants with LD higher than $0.1 R^2$ and within 1000 Kb were removed. For each GWAS, this resulted in three PRSice PRS, one for each *p*-value threshold. We also used LDPred2-Auto to train PRSs. For computational feasibility in LDPred2 we were required to limit the number of SNPs available to approximately one million, so we selected only SNPs appearing in HapMap[53]. Once weights were computed using LDPred2, we constructed the PRS using PRSice2. In a sensitivity analysis, we also stratified TOPMed-BP individuals by race/ethnic background to create separate background-specific reference panels for computing LD, and re-computed PRSice2 and LDPred2 PRSs. Similarly, in secondary analysis we evaluated the potential value in using LD reference panel that closely resemble the study population used in GWAS. To facilitate comparing effect sizes across background-specific and multi-ethnic analyses, each PRS was scaled and centered to have mean 0 and variance 1 in the TOPMed-BP dataset, computed using all individuals. Measures such as *p*-values and variance explained are not affected by rescaling.

## PRS constructions based on multiple GWAS

We constructed sums of PRSs based on different GWAS. When using independently constructed PRS, we had four PRSs based on UKBB + ICBP ($PRS_1$), MVP, ($PRS_2$), BBJ ($PRS_3$), and COGENT ($PRS_4$) GWAS; after centering and scaling each component PRS in the TOPMed-BP dataset, simple PRS sum = $PRS_1 + PRS_2 + PRS_3 + PRS_4$. Similarly, we have weighted PRS sum = $w_1 PRS_1 + w_2 PRS_2 + w_3 PRS_3 + w_4 PRS_4$. Weights were trained using the MGB Biobank, as described below. To limit the number of possible combinations, we always generated these sums using GWAS-specific PRSs that were constructed with the same approach (LDPred2-Auto, or PRSice using the same *p*-value threshold for SNP inclusion). In addition, we used the PRS-CSx software version v1.0.0 to train ancestry-specific effect sizes and create ancestry-specific PRS, and combined them as sums in the same way. PRS-CSx takes summary statistics from multiple GWAS. Each GWAS is assigned an ancestry and a reference panel that matches this ancestry; we used the provided UK Biobank reference panels. Because the software only accommodates one GWAS per ancestry, we trained two sets of PRS using PRS-CSx (PRS-CSx1, and PRS-CSx2). In both sets, we paired UKBB + ICBP with the European ancestry reference panel, and BBJ with the East Asian panel, however, for PRS-CSx1, we paired MVP with the African reference panel, while for PRS-CSx2 we paired

COGENT with the African reference panel. Each PRS-CSx training run resulted in a set of three ancestry-specific PRSs, which were combined with weighted and unweighted sums as described before for the GWAS-specific PRS. For PRS-CSx, as for LDPred2, we used only HapMap SNPs.

## Training of PRS summation weights using Mass General Brigham (MGB) Biobank

We obtained genomic data and health information for 36,434 unrelated individuals from the MGB Biobank, a biorepository of consented patient samples at MGB. Detailed methods are provided in the Supplementary Information. SBP and DBP values were medians of those recorded in the health records. In sensitivity analysis, we adjusted these values for medication use as described for TOPMed individuals. Individuals were identified as ever users of antihypertensive medications if they had records of using antihypertensive combinations, other antihypertensives, alpha or beta blockers, diuretics, peripheral vasodilators, angiotensin ii inhibitor, calcium channel blockers, or direct renin inhibitor. Age was the current age, and obesity was the current inferred status using the natural language processing algorithm of the MGB Biobank. We computed all developed PRSs in MGB Biobank participants, scaled them using mean and SDs computed in the TOPMed-BP dataset, and performed association analyses using linear models with SBP/DBP as outcomes, regressed on current age, sex, race/ethnic background, obesity status, genotypic batch, and the first 10 PCs of genetic data, as well as the PRSs to combine. PRS linear combination weights ($w_1$, …, $w_4$ or $w_1$, $w_2$, $w_3$) were extracted as the estimated regression coefficients of the respective PRSs. In primary analysis, PRS combination weights were trained using regression restricted to individuals from the race/ethnic background used for PRS evaluation in the TOPMed-BP dataset. In secondary analysis, we compared this to an approach that uses all available individuals from MGB Biobank to train the PRS combination weights, addressing the possibility that small sample sizes of non-White race/ethnic backgrounds in MGB Biobank will yield suboptimal combination weights compared to the larger sample size of the multi-ethnic, though majority White, sample.

In another secondary analysis, we investigated differences in distributions of PRSs across datasets, which can be artificially caused by differences in imputation panels and imputation quality across SNPs used, in addition to real differences in allele frequencies in the source populations[25,54]. Particularly, we assessed whether attempting to scale or "scale + match" the PRS distributions across datasets (TOPMed-BP, MGB Biobank, and UKBB Black as described later) will affect the PRS combination weights computed in MGB Biobank such that performance of PRS summations in TOPMed-BP analysis. This analysis is reported in the Supplementary Information.

## Association analysis of PRSs with BP phenotypes in the TOPMed-BP dataset

To estimate PRS effect sizes, we used linear mixed models, as implemented in the GENESIS R package[55] version v2.16.1 to estimate the association between the PRS and the corresponding trait, with relatedness modelled via a sparse kinship matrix. All models were further adjusted for sex, age, age$^2$, BMI, smoking status (ever, never, or current smoker), the first 11 genetic PCs, and combinations of study and race/ethnic background (e.g. we had ARIC-Black, ARIC-White specific intercepts when evaluating PRS association in a multi-ethnic sample) and study site when relevant. Similarly, we accounted for differences in variances across groups defined by combinations of study and self-identified race/ethnic background using a heterogeneous residual-variances model[56]. For comparability across groups and datasets, effect sizes are reported per SD of the PRSs, where SDs were computed once over all individuals from the multi-ethnic TOPMed-BP dataset.

## PRS performance measures

We examined performance of each PRS stratified by race/ethnic background. Performance was assessed by percent variance explained (PVE). PVE was computed using only unrelated individuals (selected via the pcairPartition function in the R GENESIS package) successively eliminating one individual from each pair with TOPMed kinship coefficient greater than $2^{-4.5} \approx 4.4\%$). In these individuals, PVE was calculated according to:

$$PVE = \left(1 - \frac{\hat{\sigma}^2_{prs}}{\hat{\sigma}^2_{null}}\right) \times 100\%,  \qquad (1)$$

Where $\hat{\sigma}^2_{prs}$ is the residual variance in the model that includes the covariates and the PRS, and $\hat{\sigma}^2_{null}$ is the residual variance in the model that includes only covariates. We computed confidence intervals based on 1000 bootstrap samples, using the percentile method.

## Assessment of population structure effects on alleles and PRS

As explained in the Supplementary Note 1, for each TOPMed-BP participant, we computed an estimated proportion of their alleles inherited from seven continental genetic ancestries: Europe, Africa, America, East Asia, South Asia, Middle East, and Oceania. We applied the method GAFA[49] to compute ancestry-specific allele frequencies to variants with $p$-value < 0.01, clumped with $R^2 = 0.1$ and distance = 1000 Kb, based on any of the GWAS used (SNP selection and clumping was done based on each GWAS separately). We excluded Oceania ancestry because of the small effective sample size and rescaled proportion of other ancestries so that they sum to 1 in individuals who had small fraction of Oceania ancestry. For each of the primary genetic ancestries represented in individuals in our dataset, i.e. European, African, East Asian, and American, we visualized the estimated SNP effect sizes versus effect allele ancestry-specific frequency. Combining information across SNPs, we visualized the distribution of selected PRSs across race/ethnic background group in the TOPMed-BP dataset, as well as across groups defined by continental genetic ancestry. Groups defined by genetic ancestry were made up of participants for whom at least 80% of their alleles had been assigned above to the corresponding genetic ancestry. In this visualization we included only the best-performing single-GWAS PRS and best-performing multi-PRS.

## Analysis of BP PRS associations in UKBB individuals with Black identity

To increase the number of Black participants in the dataset used for PRS evaluation, we incorporated individuals from the UK Biobank study, as described in detail in the Supplementary Information. Briefly, we used self-reported race/ethnic background (UKBB Data-Field 21000) to select 8646 individuals who self-reported as "Black or Black British" or "White and Black Caribbean" or "White and Black African" study-defined ethnic identities, which were the ethnicities that referenced Black identity. We constructed the various PRSs in UKBB Black individuals summing over the same genetic markers and weights used in the TOPMed-BP dataset. Associations of PRS with SBP and DBP in UKBB individuals were meta-analyzed with the associations estimated in TOPMed-BP Black individuals. Additional information on genetic analysis in the UKBB, including secondary analysis evaluating scaling and matching between TOPMed-BP and UKBB, is provided in the Supplementary Information.

## Validation of PRS associations and evaluation of performance across strata of BP determinants in All of Us

We used the independent All of Us dataset to further study effect estimates and variance explained by the PRSs. Variance explained, our primary performance measure, depends on phenotypic variance. As we hypothesized that some groups have larger phenotypic variances

due to environmental exposures, we studied potential differences in variance explained of PRS across different stratifications of study individuals. First, in TOPMed, we computed phenotypic variances, as well as residual variances after regressing the phenotype on baseline covariates, across race/ethnic backgrounds, and in sensitivity analysis, across groups based on self-reported race/ethnicity and further refined according to genetic similarity. Second, in All of Us, because BP phenotypes change with age and different participating studies and race/ethnic background groups may have different age distributions, we fit models that evaluate PRS effect sizes and performance by strata of age, sex, and BMI. Age strata were defined by ≤40, 40–60, and >60; sex was stratified to male and female according to sex at birth, and BMI was stratified to obesity (BMI ≥ 30) and non-obesity (BMI < 30) strata. These comparisons and analyses focused on the PRS with highest performance in the main analysis, determined by an analysis in the combined, multi-ethnic population (with adjustment to self-reported race/ethnic background). In secondary analysis we further report association analyses stratified by hypertension medication use. More details are provided in the Supplementary Information.

### PRS associations with prevalent clinical outcomes in All of Us

In a secondary analysis, we estimated the association of the best performing SBP and DBP PRSs with multiple prevalent clinical outcomes in the All of Us dataset: hypertension, type 2 diabetes, chronic kidney disease, coronary artery disease, atrial fibrillation, and heart failure, see Supplementary Information for details. The standard concept names used to define these outcomes are provided in Supplementary Table 9. Data was prepared using Python 2.7. We compared these associations to (1) our previously developed "HTN-PRS" for hypertension based on PRS summation of clump-and-threshold constructed PRS from multi-ethnic GWAS of SBP, DBP, and hypertension, as well as (2) a weighted combination of the best performing SBP and DBP PRS in the current work, with weights estimated in a regression of hypertension over the two PRS in the MGB Biobank, adjusted for current age, sex, race/ethnic background, obesity status, genotypic batch, and the first 10 PCs of genetic data. As in all other analyses, compared PRSs were standardized based on their mean and SD in the TOPMed-BP dataset. For these associations we also computed a prediction measure, the area under the receiver operating curve (AUC) using the pROC version 1.16.2 R package. Associations were estimated based on combined sample and stratified by hypertension medication use.

### Reporting summary

Further information on research design is available in the Nature Portfolio Reporting Summary linked to this article.

## Data availability

TOPMed freeze 8 WGS data and harmonized BP phenotypes are available by application to dbGaP according to the study specific accessions:

Amish: "phs000956", ARIC: "phs001211", BioMe: "phs001644", CARDIA: "phs001612", CFS: "phs000954", CHS: "phs001368", COPD-Gene: "phs000951", FHS: "phs000974", GENOA: "phs001345", GenSalt: "phs001217", HCHS/SOL: "phs001395", JHS: "phs000964", MESA: "phs001211", THRV: "phs001387", WHI: "phs001237". Summary statistics from MVP BP GWAS are available from dbGaP by application to study accession "phs001672". The summary statistics from the UKBB + ICBP BP GWAS are available at https://grasp.nhlbi.nih.gov/FullResults.aspx. The summary statistics from the COGENT BP GWAS are available at https://tarheels.live/cogentkidney/main/gwas-cogent-bp/. The summary statistics from the BBJ BP GWAS are available at http://jenger.riken.jp/en/. MGB Biobank genotyping and phenotypic data are available to Mass General Brigham investigators with required approval from the Mass General Brigham Institutional Review board (IRB).

Data from the NIH All of Us study are available via institutional data access for researchers who meet the criteria for access to confidential data. To register as a researcher with All of Us, researchers may use the following URL and complete the laid out steps: https://www.researchallofus.org/register/. Researchers can contact All of Us Researcher Workbench Support at support@researchallofus.org. Data needed to construct the selected BP PRSs generated in this study are publicly available on the Zenodo repository https://doi.org/10.5281/zenodo.7908793, and include variants, alleles, and weights for each of the PRS based on GWAS of SBP and DBP, mean and SD computed based on the TOPMed-BP dataset, and code to generate the PRS from plink files using PLINK v1.9. Data used to generate Figs. 2–5 are provided with this paper as Supplementary Datasets 1–4.

## Code availability

We provide developed scripts used to perform analyses described in the paper and code to construct the BP-PRSs in the GitHub repository https://github.com/nkurniansyah/BP_PRS, and the Zenodo repository https://doi.org/10.5281/zenodo.7908793.

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

## Acknowledgements

This study was performed as a collaboration of the NHLBI Trans-Omics in Precision Medicine (TOPMed) Consortium. We gratefully acknowledge the studies and participants who provided biological samples and data for TOPMed and CCDG. TOPMed and CCDG acknowledgements, as well as descriptions, acknowledgements, and ethics statements of

contributing studies are provided in Supplementary Note 3. TOPMed consortium researchers and their affiliations are listed in Supplementary Note 4. The views expressed in this manuscript are those of the authors and do not necessarily represent the views of the National Heart, Lung, and Blood Institute; the National Institutes of Health; or the U.S. Department of Health and Human Services. This research has been conducted using the UK Biobank Resource (application 81097). We would like to thank the participants and researchers from the UK Biobank who contributed or collected data. We thank Mass General Brigham Biobank for providing samples, genomic data, and health information data. The All of Us Research Program is supported by the National Institutes of Health, Office of the Director: Regional Medical Centers: 1 OT2 OD026549; 1 OT2 OD026554; 1 OT2 OD026557; 1 OT2 OD026556; 1 OT2 OD026550; 1 OT2 OD 026552; 1 OT2 OD026553; 1 OT2 OD026548; 1 OT2 OD026551; 1 OT2 OD026555; IAA #: AOD 16037; Federally Quali-fied Health Centers: HHSN 263201600085U; Data and Research Center: 5 U2C OD023196; Biobank: 1 U24 OD023121; The Participant Center: U24 OD023176; Participant Technology Systems Center: 1 U24 OD023163; Communications and Engagement: 3 OT2 OD023205; 3 OT2 OD023206; and Community Partners: 1 OT2 OD025277; 3 OT2 OD025315; 1 OT2 OD025337; 1 OT2 OD025276. The All of Us Research Program would not be possible without the partnership of its partici-pants. This research was supported by National Heart Lung and Blood Institute awards R01HL161012 (TS), R35HL135818 (SR), F32HL152555 (MOG).

## Author contributions

N.K. prepared PRSs and performed analyses in TOPMed, MGB Biobank, and All of Us. M.O.G. performed analyses in UK Biobank. A.T.K. harmo-nized BP phenotypes. J.W. and S.Z. derived global and local ancestry information in TOPMed. E.F., J.C.B., K.L.W., T.K., T.E., X.G., W.P., H.J.L., S.J.H., Y.G., K.Y., G.L.K., J.A.S., B.Y., S.L., S.W.S., J.E.M., X.Z., Y.D.I.C., I.T.L., C.C.G., D.M.L.J., M.F., C.K., A.C., B.M.P., C.R.I., S.S.R., R.C.K., S.R., B.D.M., N.F., D.L., J.I.R., and A.C.M. participated in either data collection, data management, or data preparation in TOPMed studies they represent. N.K., M.O.G., and T.S. drafted the manuscript. A.T.J., J.W., E.F., J.C.B., K.L.W., J.E.H., T.K., T.E., X.G., W.P., H.J.L., S.J.H., Y.G., K.Y., G.L.K., J.A.S., B.Y., S.L., S.W.S., J.E.M., X.Z., Y.D.I.C., I.T.L., C.C.G., D.M.L.J., S.Z., M.F., C.K., A.C., B.M.P., D.K.A., C.R.I., S.S.R., R.C.K., S.R., B.D.M., N.F., D.L., J.I.R., and A.C.M. critically reviewed the manuscript. T.S. supervised the work.

## Competing interests

B.M.P. serves on the Steering Committee of the Yale Open Data Access Project funded by Johnson & Johnson. All other co-authors declare no competing interests.

## Additional information

[1]Division of Sleep and Circadian Disorders, Brigham and Women's Hospital, Boston, MA, USA. [2]Department of Medicine, Harvard Medical School, Brigham and Women's Hospital, Boston, MA, USA. [3]Department of Biostatistics, University of Washington, Seattle, WA, USA. [4]Department of Biostatistics, University of Michigan, Ann Arbor, MI, USA. [5]Human Genetics Center, Department of Epidemiology, Human Genetics, and Environmental Sciences, School of Public Health, The University of Texas Health Science Center at Houston, Houston, TX, USA. [6]Cardiovascular Health Research Unit, Department of Medicine, University of Washington, Seattle, WA, USA. [7]Massachusetts Veterans Epidemiology Research and Information Center (MAVERIC), VA Boston Healthcare System, Boston, MA, USA. [8]Department of Epidemiology, Tulane University School of Public Health and Tropical Medicine, New Orleans, LA, USA. [9]Department of Public Health Sciences, University of Miami Miller School of Medicine, Miami, FL, USA. [10]Department of Pediatrics, The Institute for Trans-lational Genomics and Population Sciences, The Lundquist Institute for Biomedical Innovation at Harbor-UCLA Medical Center, Torrance, CA, USA. [11]Department of Medicine, Columbia University Medical Center, New York, NY, USA. [12]The Population Sciences Branch of the National Heart, Lung and Blood Institute, Bethesda, MD, USA. [13]The Framingham Heart Study, Framingham, MA, USA. [14]The Jackson Heart Study, University of Mississippi Medical Center, Jackson, MS, USA. [15]Department of Epidemiology, Colorado School of Public Health, Aurora, CO, USA. [16]Department of Epidemiology, University of Michigan School of Public Health, Ann Arbor, MI, USA. [17]Center for Global Cardiometabolic Health, Department of Epidemiology, Brown University, Providence, RI, USA. [18]Department of Medicine, Brown University, Providence, RI, USA. [19]Department of Pediatrics, Albert Einstein College of Medicine, Bronx, NY, USA. [20]Department of Epidemiology & Population Health, Albert Einstein College of Medicine, Bronx, NY, USA. [21]Department of Epidemiology, Harvard T.H. Chan School of Public Health, Boston, MA, USA. [22]Department of Population and Quantitative Health Sciences, School of Medicine, Case Western Reserve University, Cleveland, OH, USA. [23]Division of Endocrinology and Metabolism, Department of Internal Medicine, Taichung Veterans General Hospital, Taichung City 40705, Taiwan. [24]School of Medicine, National Yang Ming Chiao Tung University, Taipei 11221, Taiwan. [25]Division of Biostatistics, Washington University School of Medicine, St. Louis, MO, USA. [26]Department of Preventive Medicine, Northwestern University, Chicago, IL, USA. [27]Department of Psychiatry, University of Michigan, Ann Arbor, MI, USA. [28]Brown Foundation Institute of Molecular Medicine, McGovern Medical School, University of Texas Health Science Center at Houston, Houston, TX, USA. [29]Division of Public Health Sciences, Fred Hutchinson Cancer Research Center, Seattle, WA, USA. [30]Departments of Medicine and Pediatrics, University of Mississippi Medical Center, Jackson, MS, USA. [31]Department of Medicine, University of Washington, Seattle, WA, USA. [32]Department of Epidemiology, University of Washington, Seattle, WA, USA. [33]Cardiovascular Health Research Unit, University of

Washington, Seattle, WA, USA. [34]Health Systems and Population Health, University of Washington, Seattle, WA, USA. [35]Office of the Provost, University of South Carolina, Columbia, SC, USA. [36]Center for Public Health Genomics, University of Virginia School of Medicine, Charlottesville, VA, USA. [37]Department of Medicine, University of Maryland School of Medicine, Baltimore, MD, USA. [38]Department of Epidemiology, University of North Carolina, Chapel Hill, NC, USA. [39]Department of Biostatistics, Harvard T.H. Chan School of Public Health, Boston, MA, USA. [40]CardioVascular Institute, Beth Israel Deaconess Medical Center, Boston, MA, USA. [41]These authors contributed equally: Nuzulul Kurniansyah, Matthew O. Goodman. ✉e-mail: tsofer@bidmc.harvard.edu

