## [Peer Review File · Nature Communications]

Evaluating the use of blood pressure polygenic risk scores across race/ethnicity background groupsREVIEWER COMMENTS

Reviewer #1 (Remarks to the Author):

This is a manuscript I already reviewed before the update and transfer to this journal.

The results demonstrate the benefits and limitations of using PRSs for BP-related research in multiethnic populations. These are noteworthy results both in the clinical and methodological sense and the work is novel. The conclusions, the methodology, and the analyses are valid and I do not have any major additional comments over the previous ones.

Minor comments:

1. Please pay attention that the abbreviations are explained in the abstract and in the figure legends. Particularly the PRS types in the figures need to be explained.
2. Please cite doi: 10.1161/HYPERTENSIONAHA.120.16471
3. Figure 1: "withint" -> "within"

Reviewer #2 (Remarks to the Author):

This study assesses the performance and limitations of polygenic risk scores (PRS) for blood pressure phenotypes in diverse population groups. The manuscript uses several datasets for complementary evidence as well as contemporary methodology (including the recent PRS-CSx) with careful attention to avoiding overlap with the input GWASs to avoid overfitting. The study includes evaluations by sex, and for some clinical outcomes, but it is primarily a technical/methodological to support some future clinical evaluations on the topic. Key concerns about the study include the impact of including individuals on hypertension medication which could affect the generalizability of the findings, and identification of race/ethnic groups. Clarity is also needed about whether only incident cases were used for studying the impact on clinical outcomes.

Main comments:

1. My first concern related to the hypertension phenotype is about handling individuals on blood pressure-lowering therapies. It seems that in most of the test datasets, the blood pressure levels were corrected by subtracting a mmHg value from the BP measurements for persons receiving pharmacological therapy. There is nothing wrong in this for e.g. prediction of cardiovascular diseases or for running GWAS on the blood pressure phenotypes, but if we think about clinical applicability of blood pressure phenotype PRSs (one of main goals of this study, based on the intro), we would need to be careful about how these therapies impact observed predictive performance, whether it is representative of clinical care. This is because in clinical care, the potential use cases of PRSs would be quite different in individuals with and without blood-pressure lowering therapies. For instance, in those without medications, we would be interested in predicting who is likely to develop hypertension, guiding earlier initiation of blood pressure-lowering therapy, whereas in those with medications we could have a stronger focus on predicting/preventing the primary or secondary clinical outcome (e.g. major cardiovascular event). I would hope to see whether the main results are different in those with and without blood pressure-lowering therapies.
2. Another concern related to blood pressure-lowering therapy is that no medication adjustment was done in the MGB dataset. Considering that MGB is enriched in hospital cases, my concern is that using MGB as the training data for PRS summation weights without any medication adjustment could drive some of the results. It would be good to see some sensitivity analyses on whether this is the case.
3. Did the analyses on clinical outcomes include only incident cases (= cases diagnosed after

measurement of blood pressure)? Only incident cases should be used for these kind of analyses, because for risk of these diseases, blood pressure is a target of treatment and careful monitoring. Using also prevalent cases could therefore impact the associations a lot, and only incident disease cases can be used for truly understanding the impact of BP on risk of developing the diseases.

4. In the majority of analyses, ancestry was based on self-reported race/ethnic background. Firstly, how does this affect the performance of the PRSs compared to using genetically inferred ancestry (particularly in the PRS test sets)? My concern is that this would impact the performance of the PRSs particularly in other ancestries than European ancestry, and could lead to suboptimal performance and larger performance drops compared to genetically inferred ancestry. Secondly, when possible, studies tend to distinguish South and East Asian ancestry, as PRS performance usually differs quite a lot between them – did the authors look at this difference in addition to the broad “Asian” category?

5. Methods, page 14: “We computed all developed PRS in MGB Biobank participants, scaled the using mean and SDs computed in the TOPMed-BP dataset”. I understand the need for this but if I understood this correctly, the mean and SD from TOPMed-BP was used for scaling the raw PRSs in MGB. In my experience, this leads in most cases to incorrect scaling; sometimes a bit incorrect, sometimes very incorrect. The reason is that also technical factors, different imputation pipelines and imputation panels, different quality control etc. impact the raw distribution. Usually the datasets need to undergo highly similar procedures starting from pre imputation so that the genotypes of different datasets are comparable enough for this type of scaling (see PMIDs 31346163 and 32762905). My concern is that does this have a downstream effect on the analyses, considering that MGB Biobank was used for training weights for multi-PRS combinations? How large were the differences in the raw distributions between TOPMED and MGB?

Minor comments:

6. The discussion would benefit from a brief summary of what this manuscript adds to prior studies on hypertension PRSs, including the recent paper from the same team (<https://www.nature.com/articles/s41467-022-31080-2>).

7. TOPMed was used in the manuscript in several ways. If applicable, I would like to see a bit more details on how the PRS was constructed based on the WGS data in TOPMed, as genotype preparation for PRS calculation may have some quirks (see e.g. this report for Genomics England for PRSs based on their WGS data: <https://re-docs.genomicsengland.co.uk/prs/>).

8. Introduction: “- - how BP PRS may be useful for studying and quantifying the causal effects of BP on disease outcomes.” Studying causal effects based on PRSs is something that needs to be carefully done and it’s overall quite a debatable topic. Considering that causality analyses are not a part of this study, I would recommend rephrasing this to omit the word ‘causal’.

9. Introduction, “Hypertension is a major risk factor for cardiovascular disease (1,2), as well as stroke - -”. Stroke is a cardiovascular disease, so this needs to be corrected.

10. The manuscript mentions environmental factors quite often, and it is highlighted in the introduction as well. This led me to expect that the manuscript would contain results for interplay with environmental factors, which it doesn’t. I would therefore keep most mentions about environmental factors in the discussion to avoid misunderstandings about the aims of the manuscript.

11. Figures 2 and 3 would benefit from having a clearer description in the legend of what the figure depicts.

12. Discussion, first paragraph, “using varying methodologies reflective of current practice”. What does current practice refer to here (not clinical I assume)?

13. A few very minor formatting typos/clarifications:

- PVE abbreviation is missing in the abstract.

- Page 14, row 19: “First 11 genetic PCSs”. I assume the authors used 10 PCs and not 11? (based on the rest of the methods & supplement)

- Figure 1, typo in “Japanes”

- Figure 4, title: Add “race/ethnic” before the word background?

Response to review of NCOMMS-22-47874-T: “Evaluating the use of blood pressure polygenic risk scores based on largest available GWAS across race/ethnic background groups”

We would like to thank the review and the editor for reviewing and considering our manuscript, and to highlight a specific change that influence the results. Following Reviewer 2 request, we now performed a sensitivity analysis in the MGB biobank in which we raised values of SBP and DBP in individuals with records of using antihypertensive medications. Thus, we downloaded an updated MGB biobank dataset which resulted in slight changes to the main results (though patterns remained the same).

Reviewer 1:

This is a manuscript I already reviewed before the update and transfer to this journal.

The results demonstrate the benefits and limitations of using PRSs for BP-related research in multiethnic populations. These are noteworthy results both in the clinical and methodological sense and the work is novel. The conclusions, the methodology, and the analyses are valid and I do not have any major additional comments over the previous ones.

Response: Thank you for your reviews and comments!

Minor comments:

1. Please pay attention that the abbreviations are explained in the abstract and in the figure legends. Particularly the PRS types in the figures need to be explained.

Response: Thank you. We (a) fixed the inconsistency where we sometimes referred to PRS in plural as PRS and sometimes as PRSs (now we write PRSs to avoid ambiguity); (b) added “abbreviations and definitions” to figure captions to define abbreviations and PRS types.

2. Please cite doi: 10.1161/HYPERTENSIONAHA.120.16471

Response: Not citing this paper was a big oversight. We have certainly been aware of it. We now fixed this omission, and it is cited.

3. Figure 1: "withint" -> "within"

Response: Thank you, we fixed this typo.

Reviewer 2:

This study assesses the performance and limitations of polygenic risk scores (PRS) for blood pressure phenotypes in diverse population groups. The manuscript uses several datasets for complementary evidence as well as contemporary methodology (including the recent PRS-CSx) with careful attention to avoiding overlap with the input GWASs to avoid overfitting. The study includes evaluations by sex, and for some clinical outcomes, but it is primarily a technical/methodological to support some future clinical evaluations on the topic. Key concerns about the study include the impact of including individuals on hypertension medication which could affect the generalizability of the findings, and identification of race/ethnic groups. Clarity is also needed about whether only incident cases were used for studying the impact on clinical outcomes.

Response: Thank you for the thorough review and assessment. We performed additional analyses addressing your concerns as we describe below.

Main comments:

1. My first concern related to the hypertension phenotype is about handling individuals on blood pressure-lowering therapies. It seems that in most of the test datasets, the blood pressure levels were corrected by subtracting a mmHg value from the BP measurements for persons receiving pharmacological therapy. There is nothing wrong in this for e.g. prediction of cardiovascular diseases or for running GWAS on the blood pressure phenotypes, but if we think about clinical applicability of blood pressure phenotype PRSs (one of main goals of this study, based on the intro), we would need to be careful about how these therapies impact observed predictive performance, whether it is representative of clinical care. This is because in clinical care, the potential use cases of PRSs would be quite different in individuals with and without blood-pressure lowering therapies. For instance, in those without medications, we would be interested in predicting who is likely to develop hypertension, guiding earlier initiation of blood pressure-lowering therapy, whereas in those with medications we could have a stronger focus on predicting/preventing the primary or secondary clinical outcome (e.g. major cardiovascular event). I would hope to see whether the main results are different in those with and without blood pressure-lowering therapies.

Response: Thank you for this important suggestion. We now provide results stratified by use of antihypertensive medications for all multi-ethnic analyses in All of Us (i.e. in Supplementary Figures 16-17 we provide results analogous to Figure 4 and Supplementary Figures 22-23 with results analogous to Figure 5 for target BP measures, stratified by medication use); we now also provide in Supplementary Figures 25-26 secondary analyses of the prevalent clinical outcomes, stratified by medication use).

In general, BP PRS were more strongly predictive of target BP measures (assessed by PVE) in individuals who did not use antihypertensive medications. For the clinical outcomes it is difficult to assess whether associations of BP PRS with (prevalent) outcomes were stronger (in terms of statistical evidence) in one of the medication strata, due to differing sample sizes and number

of cases, which leads to differing levels of uncertainty between the two strata. It is also likely that the on-medication group differs from the off-medication group in terms of health status, lifestyle, and number of prevalent clinical CVD risk factors, and is more likely to have progressed to clinical outcomes. Therefore, comparison of the effect size of the PRS on the prevalent clinical outcomes across medication strata is not straightforward.

We wrote about this analysis in various places in the text.

Methods section, subsection “Validation of PRS association and evaluation of performance across strata of BP determinants in All of Us”, page 18 lines 13-14:

“In secondary analysis we further report association analyses stratified by hypertension medication use.”

Methods section, subsection “PRS associations with prevalent clinical outcomes in All of Us”, page 19 lines 8-9:

“Associations were estimated based on combined sample, and stratified by hypertension medication use.”

Results subsection “Estimated effect sizes and PVE of BP PRS stratified by major BP determinants in All of Us”, page 23 lines 4-8 (referring to results prior to stratification by major BP determinants):

“Supplementary Figures 16 and 17 provide the corresponding results from analyses stratified by hypertensive medication use, showing that the associations between BP PRSs and BP traits remained about the same in non-medication users as in the primary analysis, but with slightly lower effect sizes. PRS associations are weaker, but still present, in participants who use antihypertensives.”

And in the same section, referring to the analysis stratified by major BP determinants (page 23 line 21 to page 24 line 4):

“Multi-ethnic analysis stratified by both medication use and BP determinants follow similar patterns to the multi-ethnic analysis: among non-medication users (Supplementary Figure 22) the associations remain similar to the primary analysis results in Figure 5, but with somewhat attenuated effect size estimates, while in medication users (Supplementary Figure 23) the associations are weaker. Notable difference is that among medication users, BP PRS PVE is highest in the stratum of individuals ages 40 or less, unlike in those who do not use medications.”

Results subsection “BP PRS are associated with prevalent clinical outcomes in All of Us”, (page 24 line 18 to page 25 line 2):

“Results were inconsistent in analyses stratified by antihypertensive use. For example, HTN-PRS was associated with multiple outcomes (type 2 diabetes, chronic kidney disease, and heart failure) among medication users, while other PRSs were not. However, both SBP and DBP PRS and their weighted sum were (weakly) associated

with chronic kidney disease among non-medication users, while HTN-PRS was not. Given the width of the confidence intervals, we do not have sufficient power to make strong conclusions.”

Discussion section, page 29 lines 13-19:

“In secondary analysis we evaluated PRS associations with prevalent clinical outcomes in All of Us, with further stratification by antihypertensive medication use. While this analysis is useful in that it provides evidence for the potential use of BP PRSs in capturing risk that relates to specific outcomes, it is limited in that we did not evaluate associations with incident outcomes. Also, in future work, phenotypes and study samples should be further elaborated with specific, pre-defined, relevant exclusion and inclusion criteria to yield useful insights for future application in clinical care.”

2. Another concern related to blood pressure-lowering therapy is that no medication adjustment was done in the MGB dataset. Considering that MGB is enriched in hospital cases, my concern is that using MGB as the training data for PRS summation weights without any medication adjustment could drive some of the results. It would be good to see some sensitivity analyses on whether this is the case.

Response: To address your comment, we downloaded an updated dataset from the MGB biobank (which consequently led to an update of the entire analysis because the data changed), now including indicators for antihypertensive medication use. We used the same medications used in All of Us. Between 56% (Asian) and 77% (Black) have codes for using such medications which is higher than the computed percentages for hypertension status (between 19% (Asian) and 47% (Black)), where hypertension status is estimated by a chart-validated prediction algorithm. Additionally, these may not be at the same time point of the median measures of SBP and DBP which were used. Thus, we performed a sensitivity analysis in which we raised SBP and DBP values among individuals with history of antihypertensive medication use in MGB in order to recompute the weights for PRS sum to assess PVE of weighted PRS summations in TOPMed.

We addressed this comment in the manuscript.

Methods section, subsection “Training of PRS summation weights using Mass General Brigham (MGB) Biobank”, page 14 lines 4-8:

“In sensitivity analysis, we adjusted these values for medication use as described for TOPMed individuals. Individuals were identified as ever users of antihypertensive medications if they had records of using antihypertensive combinations, other antihypertensives, alpha or beta blockers, diuretics, peripheral vasodilators, angiotensin ii inhibitor, calcium channel blockers, or direct renin inhibitor.”

Results section, subsection “PRS performance by race/ethnic backgrounds”, page 21 lines 18-21:

“In another sensitivity analysis we raised the values of SBP and DBP in MGB Biobank individuals with history of antihypertensive medication use. The performance of weighted summations of PRSs in TOPMed remained similar to the analogous performance in the primary analysis (Supplementary Figure 10).”

Discussion, page 29 lines 3-7:

“However, we note that the MGB Biobank is limited in being a biobank population, that tends to have more measures related to hospital-based encounters (compared to cohort studies that have baseline and periodic surveys of all individuals). Thus, both BP phenotypes and medication data are “noisy”, and better PRS combination weights could potentially be developed.”

3. Did the analyses on clinical outcomes include only incident cases (= cases diagnosed after measurement of blood pressure)? Only incident cases should be used for these kind of analyses, because for risk of these diseases, blood pressure is a target of treatment and careful monitoring. Using also prevalent cases could therefore impact the associations a lot, and only incident disease cases can be used for truly understanding the impact of BP on risk of developing the diseases.

Response: These are only prevalent cases and are not analyzed in a model with blood pressure. We clarified this in the text by adding the word “prevalent” (e.g. methods subsection “PRS associations with prevalent clinical outcomes in All of Us” now includes the word “prevalent”, while it did not before; similarly in the corresponding results subsection). While we agree that clinically it makes sense to use PRS in a prediction context rather than “current status”, we also think that it is appropriate to perform such an analyses, especially given that this is, as we state, a secondary analysis. It is not used to explain the impact of BP on risk of developing the diseases, but rather as evidence for the potential relevance of the PRS to capture risk that relates to specific health outcomes. We note that BP PRS are fixed at birth, and that the practice of assessing genetic association with prevalent disease is the standard GWAS paradigm. We clarified the limitations of this analysis in the discussion, page 29, lines 13-20:

“In secondary analysis we evaluated PRS associations with prevalent clinical outcomes in All of Us, with further stratification by antihypertensive medication use. While this analysis is useful in that it provides evidence for the potential use of BP PRSs in capturing risk that relates to specific outcomes, it is limited in that we did not evaluate associations with incident outcomes. Also, in future work, phenotypes and study samples should be further elaborated with specific, pre-defined, relevant exclusion and inclusion criteria to yield useful insights for future application in clinical care.”

4. In the majority of analyses, ancestry was based on self-reported race/ethnic background. Firstly, how does this affect the performance of the PRSs compared to using genetically inferred ancestry (particularly in the PRS test sets)? My concern is that this would impact the performance of the PRSs particularly in other ancestries than European ancestry, and could

lead to suboptimal performance and larger performance drops compared to genetically inferred ancestry. Secondly, when possible, studies tend to distinguish South and East Asian ancestry, as PRS performance usually differs quite a lot between them – did the authors look at this difference in addition to the broad “Asian” category?

Response: We would like to clarify that we did not use self-reported race/ethnicity as an approximation to genetic ancestry (though we did fix some language specifically related to UKBB Black participants that erroneously implied that). Rather, our goal is to develop PRSs that are useful for any genetic ancestry, while assessing PRS performance within race/ethnic defined groups, which is both relevant to equity concerns about PRS performance, and acknowledges different environments that individuals from these groups may experience including potential healthcare disparities (we do acknowledge that we do not have sufficient data to describe these environments).

To your point. There are two ways that we can conceive of genetic ancestry. Coming from a perspective of thinking about admixed individuals, where an individual can have varying proportions of ancestry (we usually use continental ancestries, though finer ancestry levels can be used), focusing on groups that have a “single” predominant continental ancestry is problematic because many individuals who are admixed are left out. Even when we decide to define groups by, say, 80% or higher proportions of a specific genetic ancestry (e.g., European), it is clearly an arbitrary decision and it is not clear that the non-predominant component (e.g. non-European components) can be ignored. Notably, the TOPMed dataset has very little South Asian ancestry so we were not able to assess South Asian ancestry. In fact, all self-identified Asian individuals are of East Asian ancestry (see Supplementary Figure 3 providing distributions of continental genetic ancestry in the TOPMed dataset).

Another way to think about genetic ancestry was used in the All of Us dataset, where group labels were defined based on genetic similarity between individuals, and individuals who tend to be more similar to each other likely have similar genetic ancestry patterns, but these are not modeled explicitly.

We now report PRS performance in groups defined by genetic similarity in All of Us as a secondary analysis, but no strong conclusion can be made about PRS performance relative to our primary analysis.

We added text regarding ancestry in All of Us (which we refer to as race/ethnicity refined by genetic similarity, not “ancestry”, to distinguish between the somewhat different conceptualization/estimation of ancestry in TOPMed, where ancestries are defined at the continent level).

Methods subsection “Validation of PRS association and evaluation of performance across strata of BP determinants in All of Us” page 19 lines 5-6 (new text in bold):

“First, we computed phenotypic variances, as well as residual variances after regressing the phenotype on baseline covariates, across race/ethnic backgrounds, and in sensitivity analysis, across groups based on self-reported race/ethnicity and further refined according to genetic similarity.”

Results subsection “Estimated effect sizes and PVE of BP PRS stratified by major BP determinants in All of Us”, page 22 line 21 to page 23 line 4 (new text in bold):

“Figure 4 provides the estimated effect sizes and PVEs in each race/ethnic background, demonstrating similar pattern to the TOPMed-BP dataset, but lower PVEs, and slightly higher PRS performance in the Hispanic/Latino group compared to the White group. Supplementary Figure 15 describe similar results, from analysis stratified by groups defined by a combination of self-reported race/ethnicity and genetic similarity. Results across corresponding groups between Figure 4 and Supplementary Figure 15 are fairly similar and it is not clear that grouping individuals while accounting for genetic similarity results in improved PRS performance.”

We also explained the difference between ancestry in TOPMed and in All of Us in the Supplementary Materials subsection “Genetic ancestry in All of Us” (page 11-12):

“The All of Us study team provided ancestry labels, however, the computation of ancestry in All of Us differed from that in TOPMed, UKBB, and MGB biobank. Proportions of global ancestries were not computed, but rather an ancestry label was assigned to each participant according to their “location” in the PC space, guided by self-reported labels from the survey. Accordingly, in All of Us one of the ancestries is “AMR” referring to Latino/Admixed American, whereas in TOPMed we refer instead to the parent ancestries of Admixed Hispanics/Latinos as European, African, and Amerindian. To clarify this difference we refer to “ancestry” groups in All of Us as groups defined by a combination of self-reported race/ethnicity and genetic similarity.”

5. Methods, page 14: “We computed all developed PRS in MGB Biobank participants, scaled the using mean and SDs computed in the TOPMed-BP dataset”. I understand the need for this but if I understood this correctly, the mean and SD from TOPMed-BP was used for scaling the raw PRSs in MGB. In my experience, this leads in most cases to incorrect scaling; sometimes a bit incorrect, sometimes very incorrect. The reason is that also technical factors, different imputation pipelines and imputation panels, different quality control etc. impact the raw distribution. Usually the datasets need to undergo highly similar procedures starting from pre imputation so that the genotypes of different datasets are comparable enough for this type of scaling (see PMIDs 31346163 and 32762905). My concern is that does this have a downstream effect on the analyses, considering that MGB Biobank was used for training weights for multi-PRS combinations? How large were the differences in the raw distributions between TOPMED and MGB?

Response: Thank you for this important comment. Based on this suggestion, we visualized the distributions of PRS in MGB Biobank and in TOPMed, stratified by race/ethnic background groups, and saw that indeed they are not perfectly aligned. We added a few analyses addressing this issue: (1) the existing analysis that did not attempt to align the distributions (2) analyses that standardized the PRS separately in TOPMed and in MGB Biobank, over the multi-ethnic, combined, dataset, (3) analysis that aligned the PRS distribution between TOPMed and MGB Biobank by coordinating the PRS distributions of the self-reported White individuals (i.e. using the means and SDs based on these groups), and (4) an analysis that attempts to align PRS distribution according to genetic ancestry, where we identified groups of individuals with at least 80% continental European ancestries and used these groups to define scaling factors. We applied the same logic to the set of individuals from UKB. We next computed PRS combination weights in MGB using these rescaled PRSs and assessed PRS performance in TOPMed. All methods results in similar performance of the weighted PRS combinations and there is no clear “winner”. Therefore, the primary analysis remains as it is, and we report the analyses considering different scaling and scaling + matching approaches as secondary. We report these analyses as follows.

Methods section, subsection “Training of PRS summation weights using Mass General Brigham (MGB) Biobank”, page 15 lines 1-8:

“In another secondary analysis, we investigated differences in distributions of PRS across datasets, which can be artificially caused by differences in imputation panels and imputation quality across SNPs used, in addition to real differences in allele frequencies in the source populations (25,51). Particularly, we assess whether attempting to scale or “scale + match” the PRS distributions across datasets (TOPMed-BP, MGB Biobank, and UKBB Black as described later) will affect the PRS combination weights computed in MGB Biobank such that performance of PRS summations in TOPMed-BP analysis. This analysis is reported in the Supplementary Information.”

Results section, subsection “PRS performance by race/ethnic backgrounds”, page 21 lines 11-21:

“Example comparison of PRS distributions across datasets is provided in Supplementary Figures 6 and 7. Supplementary Figures 8 and 9 demonstrate the potential impact of PRS scaling and PRS scaling + matching approaches between the dataset used for training summation weights (MGB Biobank) and the dataset used for PRS evaluation (TOPMed-BP and UKBB Black) on PRS performance in the evaluation dataset. In brief, without explicitly matching PRS distributions across datasets, there are clear differences in their distributions between datasets. However, different scaling and scaling + matching approaches resulted in minimal differences in PVE in TOPMed-BP. In another sensitivity analysis we raised the values of SBP and DBP in MGB Biobank individuals with history of antihypertensive medication use. The performance of weighted summations of PRSs in TOPMed remained similar to the analogous performance in the primary analysis (Supplementary Figure 10).”

Minor comments:

6. The discussion would benefit from a brief summary of what this manuscript adds to prior studies on hypertension PRSs, including the recent paper from the same team (<https://www.nature.com/articles/s41467-022-31080-2>).

Response: We added the following text to the first paragraph of the discussion (page 25, lines 17-17):

“This manuscript expands upon prior work in its focus on multi-ethnic, multi-ancestry populations. Similar to our recently published paper about a multi-ethnic HTN-PRS (17), our intent was to develop PRS that are useful across individuals regardless of both their genetic make-up and their race/ethnic identity, while acknowledging that PRS performance may be related to these factors. In prior work, we considered summary statistics from multi-ethnic GWASs, trained a HTN-PRS summing trait-specific SBP, DBP, and hypertension PRS without weights, and used TOPMed as a primary dataset, and included a longitudinal and incidence analyses. Here, we developed PRSs based on multiple PRS methods and multiple GWAS summary statistics, evaluated them across population strata, and utilized multiple datasets to further assess PRS performance by strata defined by important determinants of BP.”

7. TOPMed was used in the manuscript in several ways. If applicable, I would like to see a bit more details on how the PRS was constructed based on the WGS data in TOPMed, as genotype preparation for PRS calculation may have some quirks (see e.g. this report for Genomics England for PRSs based on their WGS data: <https://re-docs.genomicsengland.co.uk/prs/>).

Response: Thank you for directing us to this link. Some of the steps reported in the link seem to be performed to create a good matching between the imputed and the sequencing data, requiring high-quality common set of variants as well as similarity in allele frequency between gnomAD and UKB, perhaps also as a quality control step, assuming that frequency should be similar if quality is high. We perform standard quality control of sequencing data, including removing SNPs that did not pass quality control and MAF-based filtering. Realizing we were not clear enough, and also that we omitted information about a requirement on missing genotypes, we revised the subsection “Quality control on summary statistics” (page 11, lines 13-16) as follows (new text in bold):

“We excluded from consideration SNPs with MAF<0.01 in the TOPMed-BP dataset and/or in the discovery GWAS (shown in Table 1), as well as SNPs that did not pass

quality control filters in TOPMed and SNPs with missing values in at least 1% of TOPMed-BP individuals. Note that we did not directly filter SNPs based on a Hardy-Weinberg Equilibrium test in TOPMed, as other quality control filters already address Mendelian inconsistencies (see link in the above subsection for quality control measures in TOPMed).”

8. Introduction: “- - how BP PRS may be useful for studying and quantifying the causal effects of BP on disease outcomes.” Studying causal effects based on PRSs is something that needs to be carefully done and it’s overall quite a debatable topic. Considering that causality analyses are not a part of this study, I would recommend rephrasing this to omit the word ‘causal’.

Response: Thank you, we completely agree with this point. We decided to delete this sentence.

9. Introduction, “ Hypertension is a major risk factor for cardiovascular disease (1,2), as well as stroke - -”. Stroke is a cardiovascular disease, so this needs to be corrected.

Response: Agreed, we removed this statement which singled out stroke from other CVD.

10. The manuscript mentions environmental factors quite often, and it is highlighted in the introduction as well. This led me to expect that the manuscript would contain results for interplay with environmental factors, which it doesn’t. I would therefore keep most mentions about environmental factors in the discussion to avoid misunderstandings about the aims of the manuscript.

Response: This is a good point. We wanted to still have some text in the introduction about gene-environment interaction because this explains the stratification by self-reported identity groups, so we deleted the last sentence of the third introduction paragraph, and replaced it. The original sentence was: “while it is important that PRS appropriately account for differences in genetic architecture across multi-ethnic populations (25), it is essential, though challenging, to disentangle genetic architecture, environmental factors, and gene-by-environment interactions”, indeed erroneously suggesting that this paper will disentangle environmental factors from genetic ones. We now have the sentence (page 6 lines 6-11):

“Thus, while PRSs may have different performance across populations due to genetic architecture factors such as LD structures, PRS associations with their phenotypes may also differ across population groups such as those defined by self-reported race/ethnicity due to various gene-environment interactions. Such environmental exposures may not be consistently measured at scale, i.e., using the same instruments across multiple studies.”

11. Figures 2 and 3 would benefit from having a clearer description in the legend of what the figure depicts.

Response: We edited the figure legends, including, we added “Abbreviations and definitions” explaining everything that is not an English word. We believe this will help readers understand what shown in the figures.

12. Discussion, first paragraph, “using varying methodologies reflective of current practice”. What does current practice refer to here (not clinical I assume)?

Response: Correct, we referred to practices of PRS development. We edited this sentence to write: “using varying methodologies reflective **of current PRS construction practices**”.

13. A few very minor formatting typos/clarifications:

Response: Thank you for identifying these and pointing them out.

- PVE abbreviation is missing in the abstract.

Response: We replaced PVE with percent variance explained in the abstract, as it was mentioned only once.

- Page 14, row 19: “First 11 genetic PCs”. I assume the authors used 10 PCs and not 11? (based on the rest of the methods & supplement)

Response: In TOPMed we indeed adjusted for the first 11 PCs. In All of Us we used 10. These are based on guidelines for the specific studies developed after assessments of population stratification captured by the PCs. We extended the title of this subsection from “Association analysis of PRS with BP phenotypes” to “Association analysis of PRS with BP phenotypes **in the TOPMed-BP dataset**”, hopefully this would be less confusing.

- Figure 1, typo in “Japanes”

Response: Fixed.

- Figure 4, title: Add “race/ethnic” before the word background?

Agreed, done.

REVIEWER COMMENTS

Reviewer #2 (Remarks to the Author):

The authors have adequately addressed my previous comments. I have only three remaining remarks:

1. When writing comment 2 below, I realized that if the effect sizes are odds ratios per SD, they are exceptionally high compared to previous studies, even up to a fairly suspicious degree. The ORs are consistently 2-4(!), even across ancestries, which is much higher than I've ever seen for blood pressure traits. Maybe I've misunderstood something and it's excellent if the authors have managed to develop such strong PRSs, but it should be kept in mind that BP trait PRSs have tended to have lower effect sizes than many other PRSs for their respective diseases (e.g. breast cancer, coronary heart disease, type 2 diabetes, with effect sizes per SD usually in the range of 1.5-2.2 per SD in European ancestry). The approaches the authors have taken for generating and testing and the PRSs are reassuring with respect to the robustness of the findings, but considering the unusually high effect sizes, I would like to see a comparison of the effect size of your novel PRSs to at least a couple of previously published PRSs for respective traits from PGS Catalog (<https://www.pgscatalog.org/>). This would reassure that the new PRS effect sizes are reliable.

2. Figure 4, Figure 5 and Supplementary figures 15-23: The x axis label says "BETA", but the legend says that the effect sizes visualized are odds ratios with 95% confidence intervals. Please clarify also for what change in PRS unit this effect size is calculated for (SD, based on the main text?).

3. Regarding the response to my previous comment number 5. The response and how the authors have addressed it is fine, but Supplementary Figure 6 and Supplementary Figure 7 seem to have the identical texts despite having different plots. What is their difference, the PRS?

Response to review of NCOMMS-22-47874A: “Evaluating the use of blood pressure polygenic risk scores based on largest available GWAS across race/ethnic background groups”

Reviewer #2 (Remarks to the Author):

Response: Thank you for your careful review. Unfortunately, we made an error in our figure legends and wrote that we report ORs in places where we actually report beta effect estimates (linear increase in outcome value). This error resulted in what appeared as the too-good-to-be-true results that you noted in your comment 1, and in the error you point out in your comment 2. Additional details below.

The authors have adequately addressed my previous comments. I have only three remaining remarks:

1. When writing comment 2 below, I realized that if the effect sizes are odds ratios per SD, they are exceptionally high compared to previous studies, even up to a fairly suspicious degree. The **ORs are consistently 2-4(!)**, even across ancestries, which is much higher than I've ever seen for blood pressure traits. Maybe I've misunderstood something and it's excellent if the authors have managed to develop such strong PRSs, but it should be kept in mind that BP trait PRSs have tended to have lower effect sizes than many other PRSs for their respective diseases (e.g. breast cancer, coronary heart disease, type 2 diabetes, with effect sizes per SD usually in the range of 1.5-2.2 per SD in European ancestry). The approaches the authors have taken for generating and testing and the PRSs are reassuring with respect to the robustness of the findings, but considering the unusually high effect sizes, I would like to see a comparison of the effect size of your novel PRSs to at least a couple of previously published PRSs for respective traits from PGS Catalog (<https://www.pgscatalog.org/>). This would reassure that the new PRS effect sizes are reliable.

Response: We apologize for making an error in the figure legends, and writing ORs instead of betas. This also refers to your comment 2. The effect sizes are indeed betas and not ORs. We report ORs only in the supplement (e.g., Supplementary Figure 24) and **the OR for hypertension is about 1.4** (depending on the specific PRS). We reviewed the recent literature reporting BP PRSs, and concluded that the effect estimates that we report are in line with other reports. Here are effect estimates that we saw in recently published papers:

Reference	Outcome	Estimated effect (beta per 1 SD)
Parchar et al. Circ Genomic and precision medicine, 2022	SBP	4.4
Parchar et al. Circ Genomic and precision medicine, 2022	DBP	2.0
Cho et al. Jama Cardiology, 2022	SBP	2.6
Cho et al. Jama Cardiology, 2022	DBP	1.5
Fujii et al. Circ Genomic and precision medicine, 2022	SBP	2.6
Fujii et al. Circ Genomic and precision medicine, 2022	DBP	2.5

2. Figure 4, Figure 5 and Supplementary figures 15-23: The x axis label says “BETA”, but the legend says that the effect sizes visualized are odds ratios with 95% confidence intervals. Please clarify also for what change in PRS unit this effect size is calculated for (SD, based on the main text?).

Response: We apologize for this error. The effect sizes that are visualized are indeed betas, and not ORs. The effect size is per SD. We clarify this by editing the figure label as follows (new text in bold):

“The figure visualizes the estimated associations (**betas in units of mmHg per 1 SD increase in the PRS, standardized according to TOPMed**) and 95% confidence intervals...”

We made this change in main Figures 4 and 5, and in multiple supplementary figures.

3. Regarding the response to my previous comment number 5. The response and how the authors have addressed it is fine, but Supplementary Figure 6 and Supplementary Figure 7 seem to have the identical texts despite having different plots. What is their difference, the PRS?

Response: Yes, the difference is that Supplementary Figure 6 visualizes the distribution of SBP PRSs, and Supplementary Figure 7 visualizes the distribution of DBP PRSs. The titles reflect that, but we now also added this in the legends (by writing “Comparison of DBP PRS distributions in...” instead of “Comparison of PRS distributions in...”).